# Polymer-Based Nanoparticles as Drug Delivery Systems for Purines of Established Importance in Medicine [note 1]

**DOI:** 10.3390/nano13192647

**Published:** 2023-09-26

**Authors:** Piotr Szyk, Beata Czarczynska-Goslinska, Dariusz T. Mlynarczyk, Barbara Ślusarska, Tomasz Kocki, Marta Ziegler-Borowska, Tomasz Goslinski

**Affiliations:** 1Chair and Department of Chemical Technology of Drugs, Poznan University of Medical Sciences, Rokietnicka 3, 60-806 Poznań, Poland; mlynarczykd@ump.edu.pl; 2Chair and Department of Pharmaceutical Technology, Poznan University of Medical Sciences, Rokietnicka 3, 60-806 Poznań, Poland; bgoslinska@ump.edu.pl; 3Department of Family and Geriatric Nursing, Faculty of Health Sciences, Medical University of Lublin, 20-081 Lublin, Poland; barbara.slusarska@umlub.pl; 4Department of Experimental and Clinical Pharmacology, Medical University of Lublin, 20-081 Lublin, Poland; tomasz.kocki@umlub.pl; 5Department of Biomedical Chemistry and Polymer Science, Faculty of Chemistry, Nicolaus Copernicus University in Torun, Gagarina 7, 87-100 Torun, Poland; martaz@umk.pl

**Keywords:** active pharmaceutical ingredients, drug delivery systems, nanoparticles, pharmaceutical technology, polymers, purines

## Abstract

Many purine derivatives are active pharmaceutical ingredients of significant importance in the therapy of autoimmune diseases, cancers, and viral infections. In many cases, their medical use is limited due to unfavorable physicochemical and pharmacokinetic properties. These problems can be overcome by the preparation of the prodrugs of purines or by combining these compounds with nanoparticles. Herein, we aim to review the scientific progress and perspectives for polymer-based nanoparticles as drug delivery systems for purines. Polymeric nanoparticles turned out to have the potential to augment antiviral and antiproliferative effects of purine derivatives by specific binding to receptors (ASGR1—liver, macrophage mannose receptor), increase in drug retention (in eye, intestines, and vagina), and permeation (intranasal to brain delivery, PEPT1 transport of acyclovir). The most significant achievements of polymer-based nanoparticles as drug delivery systems for purines were found for tenofovir disoproxil in protection against HIV, for acyclovir against HSV, for 6-mercaptopurine in prolongation of mice ALL model life, as well as for 6-thioguanine for increased efficacy of adoptively transferred T cells. Moreover, nanocarriers were able to diminish the toxic effects of acyclovir, didanosine, cladribine, tenofovir, 6-mercaptopurine, and 6-thioguanine.

## 1. Introduction

Currently, active pharmaceutical ingredients (APIs) from the purine group are used due to their antiviral and anticancer activities. 6-Thioguanine (6-TG), 6-mercaptopurine (6-MP), azathioprine pentostatin (PEN) [1,2], cladribine (CLA), clofarabine (CLO), and fludarabine (FLU) belong to important antiproliferative/anticancer purines [1,3]. However, acyclovir (ACV), valacyclovir (VACV), penciclovir (PCV) [1,2], ganciclovir (GCV), valganciclovir (VGCV) [1,4], tenofovir (TNF), didanosine (DDI), and abacavir [1,5], are among important antiviral derivatives of purine. Their mechanisms of action are similar, being based on the disruption of nucleic acid synthesis. They undergo phosphorylation and subsequent incorporation into DNA/RNA strains and, thus, terminate or slow elongation of the nucleic acid chain. Another mechanism is based on their binding to enzymes, which impacts the purine synthesis pathway or nucleic acid synthesis. While incorporations of antiproliferative purines are often nonspecific and affect both healthy and diseased cells, the activation of some antiviral purines occurs only when viral enzymes are present in the cell. For example, the first step in the activation of ACV leads to its monophosphorylated derivative and is dependent on thymidine kinase activity. The affinity of ACV towards the mammalian enzyme is 200 times lower than towards a corresponding viral enzyme. Subsequent steps are related to the phosphorylation of monophosphate to di- and triphosphate forms, which then inhibit the viral polymerase. It is worth noting that ACV triphosphate has a stronger affinity to viral than to mammalian polymerase. Similar pathways can be observed for GCV and PCV. Therefore, these substances are regarded as safe at various levels of doses and over a longer period of time. The main side effects are their nephro- and gastrointestinal toxicity [1]. Despite the fact that ACV has good antiviral activity in vitro, it may be ineffective against some viral infections. Treatment of viral hepatitis or topical administration against HSV are commonly ineffective due to insufficient concentration or retention time [6,7,8,9]. Pharmacological disadvantages, like a short half-life and low bioavailability, were eliminated by the formation of prodrugs such as VACV, VGCV, and famciclovir. Nevertheless, new analogs revealed increased toxicity on some occasions. The other antiviral purines, namely, DDI, TNF, and abacavir, are usually used against HIV infection and have different effects on human organisms than ACV, GCV, and PCV. These substances are more toxic, and therefore, more severe side effects were reported during therapy like neuropathies, pancreatitis, osteoporosis, exacerbation of hepatitis, and Fanconi syndrome [1].

The pharmacological benefits of therapy with these APIs are often limited due to low concentrations at specific sites and poor permeation through biological barriers [10,11]. 6-TG, 6-MP, and azathioprine are organosulfur compounds called thiopurines. Azathioprine is a prodrug of 6-MP, with a 1-methyl-4-nitro-1H-imidazol-5-yl substituent at the sulfhydryl group. It has a superior immunosuppressive effect and longer half-life compared to 6-MP; thus, is used to treat autoimmune diseases and to prevent organ rejection after transplantation. However, the group may be responsible for drug intolerance [1,12]. The metabolite of azathioprine, 6-MP, undergoes a further transformation into 6-TG nucleoside; hence, thiopurines share their metabolic pathway. The antiproliferative activity of 6-MP and 6-TG is more pronounced than for azathioprine, and therefore, these drugs are used in anticancer therapy, principally against blood cancers. Administration of thiopurines is associated with the occurrence of side effects such as myelosuppression, hepatotoxicity, increased risk of cancer, and opportunistic infections [1]. CLA, CLO, FLU, and PEN constitute a class of antiproliferative drugs in which purines or modified purines are connected with monosaccharides and they have halogen atoms in the structure. They are mainly used against leukemia, and their toxicity is considerably higher than thiopurines. CLA causes neutropenia in approximately 50% of patients [13]. CLO may lead to capillary leak syndrome, and all the analogs demonstrate myelosuppressive potential [1].

From the given background, there is still a need to improve the therapeutic efficacy of the APIs from the purine group. One such improvement considers the utilization of drug delivery systems (DDSs) in the form of nanoparticles. Since 2016, fifteen NP-based drugs have shown efficacy and were approved by FDA. The main advantages offered by such systems are reduced toxicity, increased bioavailability, and biocompatibility. Nanoparticles (NPs) can be easily subjected to modifications and equipped with moieties that further allow modification to biodistribution. Among the NPs, the most prominent group are polymer-based, consisting of 29% of the FDA-approved NP drugs [14]. Numerous polymers were used to prepare NPs, including poly lactic-co-glycolic acid (PLGA), chitosan, gelatin, polymethacrylates, poly(ethylene oxide) (PEO), pluronics (PLUs), poly(ethyleneimine) (PEI), poly(amidoamine) (PAMAM), and poly(propyleneimine) (PPI). These materials have advantageous properties for the preparation of NP-based drugs.

PLGA is one of the most widely used materials for NPs due to its biodegradability, biocompatibility, and the ability to adjust its physicochemical properties by modifying the ratios of components and molecular weight [15,16], similar to methacrylates. These polymers are also helpful for gastrointestinal (GI) retention, enteric coatings, pulsed release, and transdermal systems. A limited content of methacrylic acid ester with quaternary ammonium groups makes the polymers permeable [2,17,18,19,20,21,22].

Chitosan, a deacetylated derivative of chitin, gained attention as its intrinsic structure is pH-responsive—at a lower pH, it becomes less dense and subsequently releases drug molecules [23,24]. This polysaccharide has been applied in wound dressings, medical implants, drug delivery systems, and tissue engineering [25].

Gelatin is a naturally occurring polymer obtained from collagen, the properties of which depend on the type of manufacturing process. Due to its chemical characteristics, stability, and structure, including its mixture of amino acids, it has been widely applied for different purposes such as drug attachment, diagnostics, water binding, thickening, gelling, foaming, emulsifying, and as a film-forming agent [26,27,28].

Dendrimers are spheroidal particles that can be built of hyperbranched polymers such as PAMAM, PPI or PEI. Unfortunately, the toxicity of these materials is of concern, in that the surface of the dendrimers is abundant in groups that bear positive charges, which increases their toxicity [29,30,31,32]. However, modifications with acid residues can considerably reduce this disadvantage [29]. Interesting results in this field were obtained by Ziemba et al., who subjected the fourth generation of PPI dendrimers to coating with maltotriose and administered them to rats [30]. The modification meant that the toxicity was diminished and they retained the possibility of pharmaceutical activity.

Herein, we sum up the recent progress in polymer-based NPs with purine-active pharmaceutical ingredients. We believe that this approach provides an opportunity to eliminate the disadvantages of purine APIs through modifications to their physicochemical properties and pharmacokinetics. Notably, the low mean residence time (MRT) and low concentration at the site of action can be augmented by targeting moieties and bioadhesiveness, while the toxicity can be reduced by releasing drugs from a vehicle at the site. The goal is to point out milestones achieved in overcoming the limitations of purine drugs and to mark new applications of purines which have been previously registered for other indications.

## 2. Polymeric Nanospheres with Purines

### 2.1. PLGA Nanospheres

Poly(lactic-co-glycolic acid) (PLGA) is a biodegradable and biocompatible copolymer that is more and more exploited in nanotechnology. Various properties of PLGA can be adjusted by using different molecular weights and ratios of its components. Copolymers can be obtained as random or block copolymers of tunable intrinsic properties [15]. The crystalline structure of PLGA influences its mechanical resistance, swelling behavior, and capacity to hydrolyze [33]. This material is an FDA-approved polymer and reveals a wide range of erosion times and excellent water solubility. All these features contribute to its wider usage in the fabrication of drug delivery systems and tissue engineering [16,34]. PLGA has been successfully tested as a carrier for hydrophilic or hydrophobic small drug molecules or macromolecules. It prevents the transported active substances from degradation and assures their controlled release [35]. PLGA formulations have gained extensive attention due to their potential applications in the treatment of pain, inflammatory and cardiovascular diseases, cancer, and in brain imaging [33,36]. In addition, PLGA NPs are frequently modified with chitosan, PEO, or both to gain the ability to specifically interact with target sites and avoid drug burst release [36,37,38].

#### 2.1.1. PLGA Nanoparticles with 6-Mercaptopurine and 6-Thioguanine

The research conducted so far on the use of PLGA nanospheres as carriers for pharmaceutically active substances from the purine group mainly concerns analogs with antiviral and anticancer activities. One of them is 6-TG, which is an analog of the nucleic acid guanine, an oral, cell cycle-phase specific antineoplastic agent, used as a first-line treatment in acute lymphoblastic leukemia (ALL) and for remission induction and remission consolidation treatments of acute non-lymphocytic leukemias [39,40]. Furthermore, 6-TG is used for treating inflammatory bowel diseases, including ulcerative colitis and Crohn’s disease [1,41]. Previous studies indicated that 6-TG could also be a therapeutic agent for herpetic stromal keratitis treatment [42]. Another drug among thiopurines is 6-MP, which, when converted with the enzyme hypoxanthine-guanine phosphoribosyltransferase to 6-thioguanilic acid, interferes with the synthesis of guanine nucleotides by blocking the conversion of inosinic acid to adenylic and xanthylic acid. Abnormal 6-TG nucleotides are incorporated into both DNA and RNA through phosphodiester bonds, contributing to the blockade of purine nucleotide synthesis and 6-TG cytotoxicity [39,43]. The most frequent adverse reaction to 6-TG is myelosuppression. Further to the adverse effects of 6-TG treatment, hyperuricemia and hepatotoxicity, associated with vascular endothelial damage, two kinds of clinical syndromes can occur: hepatic veno-occlusive disease (hyperbilirubinemia, tender hepatomegaly, weight gain due to fluid retention, and ascites) and portal hypertension (splenomegaly, thrombocytopenia, and esophageal varices) [44,45].

Considering the limitations of 6-TG, several polymeric nanoparticles have been studied to increase the applicability of this drug. Zou et al. obtained different formulations of PLGA nanoparticles through a modified double-emulsion solvent evaporation method, where 6-MP was physically dispersed in the matrix and adsorbed on the carrier surface [46]. The optimal formulation was prepared by the addition of a solution consisting of 0.5% PVA and 6-MP to a solution of PLGA in ethyl acetate. The usage of other PLGA types (RG 502, RG 503H), dichloromethane as a solvent, and a higher concentration of poly(vinyl alcohol) (PVA) resulted in larger particles of higher polydispersity and reduced encapsulation efficiency. A cytotoxicity study on Jurkat T cells showed that the addition of 1.0 μM of the NPs revealed a lesser inhibitory impact on cell viability than the corresponding suspension of 6-MP. The IC_50_ values after 48 h of incubation were 0.36 and 1.09 μM for the suspension and the drug in the nanocarrier, respectively, and therefore, the PLGA NPs with embedded 6-MP were less cytotoxic than the free drug. In turn, the flow cytometer study provided information indicating that a more significant percentage of cancerous T cells had entered the path of apoptosis after the PLGA-6MP NPs and the suspension of 6-MP were added. Animal models (Sprague Dawley (SD) rats) provided information on the pharmacokinetics and therapeutic effects. The oral administration of PLGA-6MP did not result in a prolonged release, in contrast to in vitro studies, where a biphasic prolonged release was observed. However, after oral administration, increased C_max_, AUC, and duodenal absorption were observed for the nanocarrier with 6-MP compared to a 6-MP suspension. In addition, the administration of PLGA-6-MP NPs was associated with a lower accumulation of toxic metabolites in specific organs, like hepatotoxic 6-methylmercaptopurine in the liver and myelosuppressive 6-MP in the bone marrow. The effects listed above indicate a higher effectiveness of therapy conducted with the use of PLGA NPs. In vivo studies in rats with implanted cancer cells showed that the mean survival rate of the animals increased from 23.5 days (6-MP suspension) to 51 days (6-MP NPs). The NPs with 6-MP obtained by Zou et al. demonstrated various properties that alleviated the challenges posed by the drug. There is a possibility to increase the bioavailability, the therapeutic effect, and to reduce the amount of toxic metabolites of 6-MP [46].

Chatterjee et al. chemically conjugated 6-TG to PLGA chains through a thioester bond to improve the pharmacokinetic properties of the drug. NPs composed of the modified material were obtained by utilizing the electrospraying method, in which two steps were performed simultaneously—esterification and the formation of relatively small NPs [47]. In addition, the NPs obtained by electrospraying did not exhibit a rapid release step compared to those obtained by emulsification. Moreover, the total release was significantly extended with 60–65% and approximately 95% of the dose released within 30 and 60 days, respectively. Conversely, the NPs obtained by emulsification released approximately 92% of the API within 14 days. Taking into account the literature data on the unfavorable interaction of 6-TG with human blood albumins, the authors evaluated the process between the carrier and the protein. The strength of the affinity was assessed by recording both the changes in the diameter of the NPs and in the fluorescence spectrum of human albumins. No significant changes were observed in these parameters when the electrospray NP solution was mixed with human albumin, whereas the combination of non-conjugated 6-TG with human serum albumins resulted in a change in the fluorescence pattern. Therefore, 6-TG was retained in the matrix, and no irreversible binding of the drug to human serum protein was observed, indicating that the toxicity of the drug is preserved in the PLGA carrier. Interesting results from the studies on the PLGA NPs prepared with electrospray were obtained in in vitro studies on HeLa cells. The existence of a distinctive path of NP internalization was clearly evidenced by the change in cell viability over time. Contrary to the in vitro release study, where the release kinetics was slow, the cytotoxic effect was observed just 48 h post-addition of the NPs to the medium of HeLa cells. The IC_50_ values from the MTT assay for PLGA-6TG and free 6-TG were 12.0 and approximately 1.2 μg/mL, respectively, which means that the conjugation of the drug to the polymer chain decreased the toxicity and that the NPs reduce cell viability after 6-TG is released. Moreover, the authors stated that the drug was not released within the first 24 h outside the cells. Instead, the PLGA NPs were completely internalized through pinocytosis, and the distinctive pH of the intercellular medium was responsible for the release of the cytotoxic 6-TG. In conclusion, using the electrospray approach, the PLGA NPs with 6-TG did not reveal the burst release effect and were more stable in the extracellular environment and quickly dismantled in the intracellular environment, which was less toxic and thus helped to treat cancer [47]. The differences between the two preparation methods are shown in Figure 1.

#### 2.1.2. PLGA Nanoparticles with Acyclovir and Ganciclovir

Another active pharmaceutical agent that was embedded in polymeric NPs is ACV. It is a deoxyguanosine analog activated via phosphorylation to the triphosphate. Initially, ACV is converted to its monophosphate (ACV-MP) kinase in the body by viral thymidine. Cellular enzymes convert ACV-MP to diphosphate (ACV-DP) (cellular guanylate kinase) and ACV triphosphate (ACV-TP) (nucleoside diphosphate kinase, pyruvate kinase, creatine kinase, phosphoglycerate kinase, succinyl-CoA synthetase, phosphoenolpyruvate carboxykinase, and adenylosuccinate synthetase). ACV-TP competes with naturally occurring nucleoside triphosphates and it is incorporated into the elongating DNA chain during replication, which results in chain termination. ACV is primarily effective against actively replicating viruses; therefore, it does not eliminate the latent genome of the herpes virus. Only a small amount of ACV enters uninfected cells. Hence, the concentration of ACV-TP is 40- to 100-fold higher in HSV-infected cells than in uninfected ones. ACV-TP competitively inhibits viral DNA polymerase and, to a much lesser extent, cellular DNA polymerase. HSV resistance to ACV is associated with one of three mechanisms: mutations in viral thymidine kinase and DNA polymerase that result in the loss of thymidine kinase activity, changes in thymidine kinase substrate specificity, and reduced DNA polymerase sensitivity. ACV has been shown to treat the following infections: herpes labialis, herpes genitalis, herpes simplex encephalitis, herpes simplex keratitis, neonatal herpes infection, chickenpox (varicella), and shingles. Moreover, ACV has been reported for the treatment of acute retinal necrosis, eczema herpeticum, and oral hairy leukoplakia. Oral ACV is used to prevent HSV during the early period after transplantation in patients not receiving ACV or VGCV prophylaxis [1,39,48,49,50]. The main adverse effects rarely include nausea, diarrhea, rash, or headache, and even more rarely acute kidney damage, neurotoxicity, and local inflammation after intravenous injection [51,52]. Closely related to ACV is GCV, which acts as an acyclic guanine nucleoside analog that inhibits viral DNA polymerase [1,39]. GCV selectively inhibits viral replication. As a first step, the drug must be converted to its active form by a viral enzyme—thymidine kinase. The kinase catalyzes the phosphorylation of GCV to monophosphate; it is then converted to diphosphate by guanylate kinase and triphosphate by cellular enzymes. The triphosphate form, as a false nucleoside, is incorporated into the DNA strand, replacing multiple adenosine bases. This results in the inhibition of DNA synthesis. ACV inhibits viral DNA polymerase more effectively than cell polymerase. GCV is used for the prevention and treatment of cytomegalovirus (CMV) retinitis in AIDS patients and for the treatment of herpes keratitis and CMV prophylaxis in transplant recipients. More recent studies have focused on the potential role of ACV in the treatment of severe CMV disease, including CMV pneumonia, CMV esophagitis, and colitis, or disseminated CMV infection, in immunocompromised patients. Furthermore, previous studies have indicated that GCV has also been used against hepatitis B virus infection [1,53,54]. In in vitro studies, GCV is active against HSV-1, HSV-2, VZV, EBV, CMV, HHV-6, and HHV-8, but not against HHV-7 [55,56]. GCV is generally well tolerated, with the main adverse effect being reversible bone marrow suppression. Other adverse effects of the drug are rash, pruritus, diarrhea, nausea, vomiting, and increased serum creatinine levels and liver enzymes. Central nervous system side effects are headaches, behavioral changes, convulsions, and coma [1,57].

There is interest in polymeric NPs containing ACV and GCV. Kamel et al. obtained a series of 36 nanospheres with ACV using different combinations of surfactants (Tween 80, PVA, PLU F-127) and different types of PLGA (glycolic acid to lactic acid ratio) [58]. All of the NPs were characterized by biphasic drug release—the first burst release phase followed by prolonged, exponential release. Analysis of variance revealed that the type of polymer and surfactant used had an impact on the particle size, but no correlation between the size and concentration of the surfactant was detected. In addition, the type of surfactant affected the zeta potential. Plain PLGA NPs revealed significant negative zeta-potential, about −49.2 mV, while for those with ACV and a surfactant, −22.1 and −9.1 mV were measured. NPs with PLU F-127 in the formulation demonstrated the lowest negative potential, while those with Tween 80 demonstrated the highest. The reduction in the negative potential was due to the ionic interaction of PLGA carboxyl groups with the amine group from ACV and the covering properties of PLU. Also, the NPs where PLU F-127 was used as a surfactant were the ones with relatively small particle sizes and with faster release. Additionally, five selected formulations were coated with PLU. The coating was between 1.5 and 8.3 nm thick, yet in some cases, no coating was observed, mainly for larger NPs. The thickness of the layer depends on the hydrophobicity of the PLGA surface (the more hydrophobic the surface, the thicker the layer was due to the stronger interactions of the hydrophobic segment of the PLU) and on the molecular weight of the PLU (the larger, the thicker the layer). As a result of the coating procedure, a decrease in negative zeta potential and faster drug release was observed for these NPs. The effect of encapsulation on polymorphic changes in both the polymer and API was also analyzed. In the case of ACV, a disappearance of the transition to the crystalline state was observed, and for PLGA, a shift in the glass transition temperature towards higher values was observed. The most advantageous variant considered was NPs prepared using PLGA with a ratio of mers—lactic acid:glycolic acid of 75:25 and a 1% solution of PLU F-127 as surfactant, and without coating. The use of this combination in vivo on rabbits resulted in a twenty-nine-fold increase in the mean residence time of the API in the body and over thirty-fold longer biological half-life compared to a commercial drug formulation. A comparison of the pharmacokinetic parameters is presented in Table 1. Moreover, during parenteral administration of the PLGA NPs with embedded ACV, no thrombophlebitis or inflammation was observed, while the commercial drug was associated with these adverse effects. To sum up, the presence of PLU in the entire volume of the nanostructure contributed to a reduction in NP uptake by the phagocytic system and extended blood circulation of the drug [58].

A comparable carrier created by Jain et al. [6] (described later in the work) was produced by Gupta et al. [9]. PLGA chains were conjugated with galactose via an ethane-1,2-diamine linker to specifically target liver tissue. The conjugation of galactose resulted in an increase in the NPs’ diameters and a reduction in the cumulative release of ACV. The sizes of the NPs were 173 and 198.1 nm for gal-NP-ACV and NP-ACV, respectively, and the polydispersity index (PDI) was almost at the same level, 0.107 and 0.102 for gal-NP-ACV and NP-ACV, respectively. The non-galactosylated NPs had a more negative zeta potential, with a value of −20.8 mV, over two-fold lower than galactosylated NPs (−8.5 mV). Both NPs were characterized by a biphasic in vitro release with an initial burst release of up to 4 h. The cumulative release over 168 h was 94.01% for NP-ACV and 73.12% for gal-NP-ACV. Modifying the carboxyl groups of PLGA and the chemical bonding of galactose to the chain positively affected the accumulation of the drug in the liver and reduced the concentration of API in kidneys, the organ responsible for most of the excreted dose of API, which was studied in mice. The concentration in the tissue for gal-NP-ACV was 1207.93 μg/g, while for the plain drug it was 124.30 μg/g. Interestingly, an augmentation of the concentration in the liver was also observed when non-galactosylated NP-ACV was administered. In addition, the introduced modification significantly reduced the hemolytic effect resulting from the carrier itself and the drug. Moreover, other pharmacokinetic parameters were improved when administered to mice—t_1/2_ increased from 3.406 h to 23.13 h, MRT from 4.85 to 24.42 h, and AUC_0–∞_ from 238.246 to 428.50 μg·h/mL [9]. The idea for targeting liver tissue is shown in Figure 2.

Alkholief et al. developed a system for drug delivery to the eye for ACV by employing PLGA NPs with the addition of a stabilizer—D-α-tocopheryl polyethylene glycol 1000 succinate (synthetic, pegylated vitamin E) or polyvinylpyrrolidone (PVP) [7]. Using synthetic vitamin E with PVP contributed to a positive value of the zeta potential, resulting in better adherence of the NPs to negatively charged eyeball tissues. The obtained NPs were tested for compatibility with the precorneal tear film. Specifically, the pH value was about 6.88 after synthesis and 7.21 after 30 days (optimal pH value 6.8–7.4), the surface tension was 41.28 mN·m^−1^ (lachrymal fluid 40–50 mN·m^−1^), and the refractive index was between 1.353 and 1.367 in the tested period (lachrymal/tear fluid 1.34–1.36). The initial viscosity of the NPs was about 15 mPa·s higher than the optimal value (20 mPa·s), nevertheless it was necessary for corneal and precorneal retention and, thus, the value was acceptable. Additionally, based on literature data, it was not found that the viscosity value could cause temporary impairment of vision. Therefore, as the pH value, refractive index, and surface tension of the NP solution were close to those of the tear/lachrymal fluid, the compatibility and easy mixing with the fluids was confirmed. In vitro release studies have shown that ACV was released in three phases: rapid release (8 h), controlled release (for the next 6 h), and exponential release (for a further 58 h) [7]. The time dependence of the released dose of ACV from the PLGA-TPGS-ACV best correlated with the Korsmeyer–Peppas model, that describes release from polymer systems [7,59]. In vivo studies on rabbits indicated the irradiation ability of the PLGA-TPGS-ACV at the application site, along with a pharmacokinetic evaluation. There were no significant deviations from the normal functioning of rabbit eyes; only mild redness and mucoidal discharge in one rabbit was documented. Therefore, it was concluded that the proposed drug delivery system had adequate safety. In the case of tear fluid taken from rabbits treated with PLGA-TPGS-ACV, the appropriate concentration of the active substance was found to have been maintained up to 24 h after administration, which is a significant advantage of the vehicle, as for the ACV solution, the active substance was not quantifiable 6 h post-administration. Additionally, t_1/2_ was extended from 1.908 h to 3.264 h and MRT from 3.072 h to 6.771 h, as compared to the aqueous solution of ACV. Therefore, the designed formulation overcomes the defect of ACV being associated with a short residence time in the tissues of the organ of vision [7]. The significant difference between PLGA-TPGS and PLGA-PVP is shown in Figure 3.

Jwala et al. developed a thermosensitive gel based on PLGA-PEG-PLGA with PLGA NPs for the ocular administration of dipeptide derivatives of ACV [8]. The formulations consisted of the PLGA-based carrier and an ACV prodrug (one of the possible variants of the set consisting of L-valine and D-valine). The prodrugs (except for those with two D-valine molecules) revealed an affinity to peptide transporter protein 1, widely expressed on the cornea, thus sufficiently undergoing bioconversion to ACV (faster than the biological half-life of the compound). The derivatives with the D-valine moiety at the end of the dipeptide chain were correlated with higher stability, whereas those with L-valine at the same site were correlated with faster degradation to ACV. Moreover, docking results obtained in the in silico study indicated that analogs with L-valine had a better affinity to the peptide transporter and only D-valine-D-valine-ACV (DD-ACV) was not able to inhibit the uptake of another ligand ([3H]-glycylsarcosine) of the transporter in vitro. Additionally, the derivatization resulted in similar or lower cytotoxicity towards rabbit primary corneal epithelial cells (rPCECs). Based on the fact that L-valine-L-valine-ACV (LL-ACV) and L-valine-D-valine-ACV (LD-ACV) were the two compounds with the fastest bioreversion the best uptake by rPCECs and revealed no cytotoxicity in tested concentrations, the two analogs were evaluated further. The most compatible grades of PLGA were PLGA 65:35 and PLGA 75:25 for LD-ACV and LL-ACV, respectively, as the entrapment efficiencies and drug contents were the highest among the groups. Both obtained types of NPs had a diameter under 200 nm and were characterized by a bimodal release pattern, with burst release as the first phase. However, when the NPs were dispersed in PLGA-PEO-PLGA thermosensitive gel (gel temperature 32–60 °C), the bimodal release pattern changed and the burst release was eliminated. The formulation had the potential for use in the treatment of eye infections caused by the herpes simplex virus due to the increased average residence time in the desired place and permeation to the appropriate tissues associated with affinity to a specific receptor. However, further studies in this direction are required to confirm the therapeutic effect in vivo [8].

Bhosale et al. reported on mucoadhesive NPs for ACV oral delivery, consisting of PLGA, polycarbophil, and PLU F-68 [60]. The influence of different concentrations of applied materials on particle size, drug entrapment, and drug release was evaluated using 2^3^ factorial design. The authors employed a solvent deposition method for the particle preparation, where an organic solution of PLGA was poured into an aqueous solution of ACV, PLU F-68, and polycarbophil. The increase in the PLGA and polycarbophil concentrations caused the increase in the mentioned dependent variables, while the increase in PLU F-68 resulted in a drop in the values. Next, the mucoadhesion of the NPs was studied. The experiment was carried out on a mouse duodenum and jejunum by reflecting the normal function of these organs. The most mucoadhesive NPs were those with greater amounts of polycarbophil, a mucoadhesive polymer. The formulation, which was prepared by usage of 175 mg PLGA, 0.30% PLU F-68, and 0.10% polycarbophil, presented the greatest mucoadhesion, with 67.3% of the NPs remaining. Thus, the results indicate the possibility of improving ACV therapy by using the formulation proposed by the authors [60]. Thanks to the increased retention, it was possible to reduce the frequency of dose administration, and also the toxicity of ACV towards the digestive system. It is worth noting that this approach allows for the elimination of the use of VACV—a prodrug of more significant toxicity towards the digestive system [61,62].

PLGA nanospheres with ACV were embedded in a mucoadhesive buccal film by B. Al-Dhubiab et al. [63]. The prepared formulations were characterized by a size of 190–700 nm (average 400 nm), a zeta potential of approximately 28.73 mV, and PDI of 0.16. During the study, a film with adequate hydration, release, mucosal adhesion strength, and ex vivo permeation through the membrane was selected. In in vivo comparative studies on rabbits, the ACV-loaded NPs in films were found to have more favorable pharmacokinetics than free ACV administered orally, as a three times higher maximum concentration in blood and nearly eight times higher AUC were achieved compared to oral administration. In addition, after removal of the films from the oral cavity 4 h post-administration, an increase in the ACV concentration in the blood was still observable for a subsequent 2 h, which may have been caused by permeation of nanospheres through the oral mucosa into the circulatory system [63]. Therefore, the chosen formulation could be a promising alternative to administration through the digestive system/parenterally.

Ensign et al. developed PEO-coated PLGA NPs for intravaginal administration to act as protection against HSV [10]. In the work, the authors compared the developed NPs with a standard gel and conventional polymer NPs. PLGA NPs coated with PEO were characterized by a higher ability to penetrate through vaginal mucus, and their retention was much higher—60% of the NPs remained after 6 h in the genital tract of mice. At the same time, only 10% of the applied conventional NPs remained. Next, studies evaluating the safety of the NPs were conducted and compared to a positive control (nonoxynol-9), a TNF gel used in a clinical trial, and a hydroxyethylcellulose placebo gel. There was no increased release of cytokines during repeated administration of the developed NPs, and no abnormalities were detected as compared to the placebo group in mice, while in the positive control, side effects appeared. This is favorable, as abnormalities in these parameters could possibly increase the likelihood of infections [10]. Additionally, the protective properties of other developed mucus-penetrating particles that contained ACV were evaluated in the study. These consisted of PLU, ACV monophosphate, and the chelating agent zinc acetate. The formulations were administered vaginally 30 min prior to the application of the HSV-2 strain in a dose that normally should infect around 85% of mice subjected to study. In the group of mice receiving the developed formulation, 53.3% of the population was protected against HSV infection, while in the group receiving the soluble drug, 16% were protected. These results indicated the possibility of using the formulation for protection against HSV infection. Overall, the developed formulation increases the protective effectiveness of ACV administered vaginally against HSV and eliminates the disadvantages of current formulations associated with the rapid removal of the API from the vagina.

#### 2.1.3. PLGA Nanoparticles with Tenofovir and Its Prodrugs

There are also studies concerning TNF, which is an acyclic nucleotide diester analog of adenosine monophosphate. Since 2008, TNF has been available in two oral forms: tenofovir disoproxil (TDF) and tenofovir alafenamide (TNFA) [39,64]. TDF is hydrolyzed to TNF, which is phosphorylated by cellular kinases to the active metabolite, tenofovir diphosphate. The disphosphate inhibits viral polymerases by directly competing for a binding site with the natural deoxyribonucleotide substrate and, once incorporated into DNA, by terminating the DNA chain. The mechanism of action of TNFA is very similar: the drug is in plasma all the time in the form of a prodrug, and then it enters cells and is converted to its active form, TNF. This significantly reduces its nephrotoxicity. TNF-DP exhibits broad-spectrum activity against viral DNA polymerases, while it has a low affinity to human DNA polymerases α, β, and γ, which is associated with its selective toxicity [1,39,65,66]. TNF has been used for the treatment of HIV-1 infection in combination with other antiretrovirals and is the preferred drug for the treatment of hepatitis B infection [67]. In general, the drug is well tolerated. Side effects include mild to moderate gastrointestinal disorders, such as nausea. In addition, the drug can cause renal failure and rare cases of Fanconi syndrome, proximal tubule dysfunction. Scientific reports indicate the occurrence of bone fractures in HIV-infected patients during TNF therapy. TNF may be associated with rebound HBV replication and the exacerbation of hepatitis. Moreover, TNF reduces bone mineral density, which can lead to osteoporosis and hypophosphatemic osteomalacia. In addition, studies in animal models have found that TDF reduced bone mineral density [68,69,70,71].

PLGA NPs with TNF or its prodrug have been studied as a potential drug delivery system in pre-exposure prophylaxis [72,73,74]. Zhang et al. developed semen-responsive NPs (PLGA:PMAA in different ratios) with embedded TNF or TDF for vaginal administration [72]. The prepared NPs had a diameter of 250 nm and were able to encapsulate 16.1% and 37.2% of TNF and TDF, respectively. The responsive character of the NPs was due to different pH values between the vagina (pH 4–5) and sperm (pH 7.5) and increased solubility of the NP matrix consisting of a copolymer of methacrylic acid and methyl methacrylate and PLGA in a ratio of 75:25 at higher pH values. With this composition, 78.5% and 19.6% of the dose were released within 72 h at pH 7.6 and 4.2, respectively. In addition, the cytotoxicity of the NPs was investigated on female genital epithelial cells. Based on these studies, cytotoxicity was not observed up to an NP concentration of 10 mg/cm^3^ [72]. Cunha-Reis et al. prepared PLGA-NPs with TNF and embedded them in a polymeric film that contained free TNF in order to enhance retention of the drugs in the vagina [73]. The in vivo studies in mice determined that 80% of the administered dose of NPs in solution (PBS) was removed from the vagina within 30 min. Compared to the NPs in film, an improvement of seven percentage points was observed. These results are not satisfactory, although they may indicate a potential development path for vaginal formulation [73]. Machado et al. also pointed out the problem of the rapid removal of NPs from the vagina; thus, the NPs were dispersed in PVA-HPMC films [74]. In the paper, the authors also focused on increasing the TNF concentrations in the NPs, as it was low due to TNF’s hydrophilic nature. For this purpose, the NPs were enriched with octadecyl-1-amine, which, furthermore, increased toxicity in vitro. However, in in vivo studies on mice, no increased toxicity was observed. Moreover, the addition of octadecyl-1-amine caused a change in the zeta potential from negative (−18.7 mV) to positive (48.4 mV), and a higher value of PDI was observed, but no significant change in the particle size was noted. With this vehicle, 80% of the dose was released in the first 5–10 min in an in vitro release study [74]. Overall, the authors stated that the DDS has the potential to act as alternative for gels and films, however, further studies are needed. Another study, by Destache et al., focused on PLGA NPs contained in a thermosensitive gel with the prodrug of TNF [75]. In order to increase the prodrug content in the NP, sodium deoxycholate was used, which formed complexes with TDF, thus reducing its hydrophilicity and increasing its affinity towards PLGA. The NPs were obtained through the O/W emulsion method and were characterized by a diameter size of 148.6 nm, surface charge of −26.7 mV, and encapsulation efficiency of 52.9% (without the complex formation the EE was only 16.1%). Studies on humanized mice exposed to HIV-1 have shown the effectiveness of NPs in a thermosensitive gel, as none of the animals became infected in the group where the formulation was administered vaginally, contrary to the control group. In addition, the 1% TDF NPs were found to have an increased retention capacity in the mice’s genital tracts, as the average concentration value of the drug 24 h post-administration was almost twice as high as after administration of a normal gel (1% TNF). At the same time, the authors pointed out that the difference is not significant and the study should be expanded [75].

PLGA NPs have also been studied for the potential of using them in a highly active antiretroviral therapy [76,77]. Belletti et al. prepared NPs targeting one of HIV’s permissive cells—macrophages [76]. The concept underlying the passive mechanism of action of NPs, where opsonization and subsequent recognition followed by uptake by cells of the phagocytic system was proposed. Typically, this mechanism is responsible for reducing the therapeutic efficiency of NP-based DDS. In that situation, uptake by those cells is desirable since they are the therapeutic target. Chitosan and PLGA were used to prepare NPs, which translated into increased encapsulation of the active substance compared to NPs containing only chitosan or PLGA. In addition, a protective layer of chitosan formed on the surface of the NPs equipped them with a positive charge and an average diameter size of 400 nm. In vitro release studies using a dialysis technique at pH 7.4 showed that TNF was rapidly released from the carrier, and its release profile was similar to that of the active substance alone, but at pH 4.6, a difference was observed. Therefore, the NPs did not significantly modulate the release profile at pH 7.4 [76]. Shailender et al. prepared oral PLGA NPs with TDF using the Box–Behnken design model [77]. The obtained NPs presented a zeta potential near neutral (−4.8 mV), but this did not adversely affect their stability due to the occurrence of steric stabilization. Other parameters, size, PDI, and encapsulation efficiency, were 218 nm, 0.23, and 57.3%, respectively. In vivo studies on mice showed that the prepared NPs significantly improved pharmacokinetics during oral administration compared to free TDF. A six-fold increase in the value of the AUC and a two-fold increase in the biological half-life was observed. At the same time, no additional toxicity was observed for NPs with the active substance in relation to the administration of the API alone. The designed vehicle also provided protection of the prodrug against metabolic degradation in the intestine and the uptake took place by clathrin-mediated endocytosis, which was supported by ex vivo everted gut sac studies in rats [77].

To sum up, the major benefits of using purines in PLGA NPs are presented in Figure 4 and Table 2.

### 2.2. Methacrylate Copolymer-Based Nanospheres

Polymethacrylates are synthetic cationic and anionic polymers of dimethylaminoethyl methacrylates, methacrylic acid, or methacrylic acid esters, with varying properties depending on their pendent functional ester groups [78]. The most popular methacrylate copolymers for pharmaceutical purposes have been supplied under the trade name Eudragit^®^ and were first introduced in 1954 as Eudragit L and Eudragit S for enteric coating. As the development of Eudragits progressed, the ratio of monomers of different functional groups changed and led to the design of coatings for rapid disintegration and sustained release formulations. The production of various grades of Eudragit contributed to the development of novel formulations and allowed for their application not only as a film coating but also in granulation, direct compression, and melt extrusion. Eudragit RL in combination with Eudragit RS is mainly intended for matrix structures and customized release profiles, e.g., sustained drug release in products such as NPs, mucoadhesive tablets and patches, solid dispersions [2,17,18,19,20,21,22]. When considering Eudragit RSPO, it is insoluble and reveals low permeability with pH-independent swelling. According to many reports, Eudragit RSPO is mainly used in the sustained drug release formulations of microballoons, tablets, microspheres, and microtablets [79,80,81,82,83].

Methacrylate-based nanospheres have been so far studied for their use as a carrier of ACV, as compiled in Table 3. Gandhi et al. developed NPs with prolonged release of ACV, based on copolymers of methacrylates, including those containing quaternary nitrogen atoms [20]. The work focused on the development, preparation, and determination of NPs with the most beneficial physicochemical properties. The most promising formulations revealed had a drug to polymer ratio of 1:2 and had 0.25% *w/v* of PLU 68 as a stabilizer. The NPs were characterized by a diameter of 326 nm, PDI of 0.694, drug entrapment efficiency of 79.34%, and relatively rapid release of the active substance in vitro (over 75% after 24 h). In addition, the remaining results indicated that ACV was presented in an amorphous state and that there were no strong bonds between the drug and the polymer matrix [20]. Ramyadevi et al. also developed NPs with ACV based on copolymers of methacrylates, specifically those containing quaternary nitrogen atoms and PVP, and incorporated them into a PLU gel to act as a vaginal DDS [84]. The formulation was tested on rats for the possibility of use in the prevention of sexually transmitted HSV infections. Compared to ethylcellulose NPs, methacrylate-PVP NPs had a higher capacity for permeation through the vaginal membrane and had a more favorable ACV concentration profile over time at the administration site compared to a regular ACV solution. Moreover, polymethacrylate-PVP NPs enhanced the bioavailability of ACV approximately two-fold, and prolonged the time for which ACV was at a specific concentration (0.4–0.8 μg/mL, without a sharp peak after administration), which may be associated with higher drug retention in the vagina. Possible toxicity was also explored. In vitro studies on vaginal epithelial cells and histological studies of rats’ organs were performed. In neither study was there any changes that indicated a more toxic character of the NPs compared to an ACV solution of the same dose [84].

## 3. Nanogels with Purines

### 3.1. Chitosan-Based Nanogels

Chitosan is a deacetylated derivative of the linear polysaccharide chitin [24]. Chitin occurs abundantly in crustacean shells, insect cuticles, algae, and in the cell walls of fungi [85]. During the synthesis of chitosan, some N-acetylglucosamine moieties are converted into glucosamine units, and many protonated -NH_2_ groups present in the chitosan structure contribute to its solubility in aqueous acidic media [24]. The number of deacetylated groups in chitosan is determined by the so-called degree of deacetylation [86]. However, it is still poorly soluble at neutral or basic pH, which has been attributed to the high crystallinity caused by hydrogen bonds and acetamido groups present in its structure [87]. Although it has high viscosity and a tendency to coagulate proteins at high pH [88], it has received increasing attention in recent years due to its numerous advantages, such as its film-forming ability, biodegradability, biocompatibility, antimicrobial activity (fungi, bacteria, viruses), antiacid, antiulcer, and antitumoral properties [89]. Several attempts have been made to improve the properties of chitosan derivatives, which depend on the degree of deacetylation, molecular weight (MW), and the distribution of acetyl groups along the main chain [25,90,91]. The modifications of chitosan were focused mainly on the transformation or substitution of the OH and amine (NH_2_) groups [25].

Chitosan NPs constitute one of the most studied carriers for purine drugs. 6-MP, 6-TG, CLA, ACV, and DDI have been combined so far with chitosan. A summary of the use of nanogels as DDSs for drugs from the purine group is presented in Table 4. Govindappa et al. studied the toxicity of chitosan NPs with 6-MP and compared it to a free drug using Wistar rats as model organisms [92]. The significant findings are shown in Figure 5. The preparation (gelation technique) was optimized to yield NPs with an average size of 187 nm, PDI of 0.462, zeta potential of 54.9 mV, and loading efficiency of 25.23%. In oral acute toxicity studies, high doses of 6-MP and corresponding amounts of chitosan NPs with 6-MP were administered. At 300 mg/kg, none of the three animals treated with the MPs died, while in the plain drug group, one individual died. At a higher dose, 2000 mg/kg, two animals died in the NP group and all three in the free 6-MP group. These results indicate that the lethal dose of chitosan NPs of 6-MP is higher than the dose of the plain drug. The next study evaluated subacute toxicity after a treatment cycle of 4 weeks. Biochemical and hematological parameters presented more favorable values in the group receiving chitosan NPs with 6-MP than the pure solution of the drug, both at high doses. Histopathological examinations of the kidneys and liver were also performed. In both cases, deviations from the normal state were found; however, the authors hypothesized that due to the prolonged release, the NPs were less toxic to the liver due to a lower accumulation of toxic metabolites [92].

A related study concerned the preparation of chitosan NPs from 6-MP and the determination of their basic pharmacokinetic parameters [93]. The NPs were prepared using the ionic technique, utilizing sodium tripolyphosphate as a cross-linking agent. Among the obtained NPs, those with a chitosan to sodium tripolyphosphate ratio of 0.50:0.50 mg/mL were the most promising, as their size was the lowest (137.9 nm) and loading efficiency the highest (29.10%). Moreover, their PDI and zeta potential were also acceptable, with values of 0.239 and 53.8 mV, respectively. Therefore, this formulation was chosen for further studies. An in vitro release study showed that 6-MP dissolved faster in a more acidic acceptor solution. Namely, 95% and approximately 77% of 6-MP was released within the first 24 h at pH values of 4.8 and 7.4, respectively. In vivo pharmacokinetic studies on rats after oral administration of free 6-MP and 6-MP embedded in chitosan NPs revealed an increased absorption time in the GI tract (mean absorption time from 3.85 h to 7.98 h). Moreover, a higher bioavailability was observed, with a value of 56.52% for chitosan NPs with 6-MP, which was about 19 percentage points higher than for the free drug. In the next part of the study, an in vitro evaluation of cell cultures was conducted. Cytotoxicity determination on HT-1080 and MCF-7 cells showed no significant difference in cell viability between treatment with 6-MP and 6-MP in chitosan NPs, as the IC_50_ values were 6.14 and 6.44 μM for free 6-MP against HT-1080 and MCF-7, respectively, while for 6-MP embedded in the chitosan NPs they were 6.74 and 6.85 μM for HT-1080 and MCF-7, respectively. However, the DDS contributed less to the formation of reactive oxygen species, and in the MCF-7 cell group, it was found that 6-MP in NPs increased the percentage of cells exhibiting early apoptotic features [93]. Another study of a chitosan-based carrier was conducted by Rajashekaraiah et al. [23]. The authors developed a platform for the delivery of 6-TG in combination with curcumin, a substance with potential anticancer activity. The size, PDI, zeta potential, and entrapment efficiency of these NPs prepared by the ionic gelation cross-linking method were 261.63 nm, 0.34, 15.97 mV, and 44.27%, respectively. The obtained NPs were characterized by a pH-dependent release, with the active substance released faster at a lower pH due to the looser organization of chitosan chains, and slower in more basic solutions, where chitosan chains were packed tighter. After 4 h, 57.14% and 19.71% were released at pH values of 4.8 and 7.4, respectively. In vitro studies on ovarian cancer cells showed that the nanocarrier with 6-TG and curcumin revealed a higher antitumor effect than the 6-TG or 6-TG NPs used alone without curcumin. The IC_50_ values for the PA-1 cell line were 5.81 and 3.92 μM for free 6-TG and in the nanocarrier, respectively. On MCF-7, the IC_50_ values were 23.09 and 17.82 μM for free 6-TG and in the nanocarrier, respectively [23].

In the past decade, two research teams have addressed the issue of chitosan-based nanoformulation for CLA. CLA, a chlorinated analog of deoxyadenosine, in a sequential three-step process, undergoes intracellular phosphorylation by deoxycytidine kinase and conversion to its active form 2-chlorodeoxyadenosine triphosphate (2-CLA-ATP), then the drug is incorporated into DNA. This action causes DNA strand breaks and depletion of NAD and ATP, inducing apoptosis and inhibition of reductase [1,39,94]. CLA is approved for the treatment of active hairy cell leukemia (leukemic reticuloendotheliosis), as defined by clinically significant anemia, neutropenia, thrombocytopenia, or disease-related symptoms. Furthermore, CLA is used as an alternative agent for the treatment of chronic lymphocytic leukemia, low-grade non-Hodgkin’s lymphoma, and cutaneous T-cell lymphoma, relapsing multiple sclerosis. The main adverse effect of CLA is bone marrow suppression. Moreover, therapy with CLA causes nausea, infections, high fever, headache, fatigue, skin rashes, and tumor lysis syndrome. Bacterial, viral, and fungal infections have also been diagnosed in patients treated with the intravenous form of CLA [95,96,97]. Domaratzki and Ghanem focused on the production of the carrier and the assessment of the basic properties of the obtained material [98], while Nasrabadi et al. focused on the thermodynamic and chemical aspects of the carrier [99]. Domaratzki and Ghanem obtained NPs that were in the range of 636–820 nm, depending on the drug concentration [98]. Interestingly, compared to unloaded NPs (923 nm), their size was smaller, which could indicate that CLA increases the strength of ionic bonds or/and contributes to the number of positive sites available for sodium tripolyphosphate. The obtained NPs were characterized by a biphasic release; in the first phase, that lasted 6 h, 38% of the CLA was released, followed by a period from 24 to 34 h without significant release. After this time, a further release occurred, which lasted up to 80 h from the start of the study until total release was achieved [98].

An interesting study on the targeted delivery of ACV to hepatocytes was conducted by Jain et al. [6]. Lactose was connected to succinate-modified surface chains of chitosan, in order to increase the accumulation of the active substance in the liver by internalization of the NPs through endocytosis mediated by asialoglycoprotein receptors. The visualization of the NPs is shown in Figure 6. The particle size, PDI, zeta potential, and encapsulation efficiency were measured (Table 4). The conjugation of lactose to the surface decreased the positive zeta potential, which initially was 7.3 mV. In addition, the vehicle was characterized by prolonged release in vitro; at pH 7.4 after 72 h, the amount of the released dose was 27.3%. This was slower compared to non-lactosylated particles, where the release was about 56.1% over the same time, indicating that lactose created a protective coat on the surface of particles. The authors also presented in vivo results on mice. The lactosylated NPs with ACV significantly prolonged the mean residence time compared to free ACV and non-lactosylated NPs. The MRT values were 19.67, 2.91, and 8.31 h, respectively. Also, the AUC values over different time intervals, t_1/2_, and clearance indicated that the lactose-coated NPs had more favorable pharmacokinetic properties than the two evaluated formulations. Moreover, a biodistribution study indicated that lactosylated NPs preferentially accumulated in the liver and revealed a reduced entry to other organs. The percentage of dose recovered after 8 h was significantly higher for non-lactosylated and lactosylated NPs than for free ACV. Moreover, a fluorescence microscopy study of mouse liver tissue revealed that the lactosylated NPs accumulated more in the hepatocyte tissue sac than two other compared formulations [6]. It is worth noting that the studies of Gupta et al. [9] and Jain et al. [6] provided guidelines for targeting liver tissue through the ASGR1 receptor and initially evaluated the feasibility of using this receptor as a target for drug delivery systems [6,9].

Another direction for the development of chitosan carriers for ACV concerned their use as eye drops. NPs can have a beneficial effect on the residence time of the drug in the organ of vision, which is usually short for ACV, through adhesion or electrostatic interactions with the cornea and conjunctiva. Rajednran et al. optimized the procedure (ionic gelation) for the preparation of chitosan NPs with ACV. For the release study, the authors selected NPs with the smallest diameter of 200 nm, zeta potential of 36.7 mV, PDI of 0.16, and the greatest loading capacity of 25% [100]. NPs were characterized by a biphasic release: 52.46% in the first 12 h and up to 90.70% in the next 12 h [100]. The obtained NPs revealed the potential to increase the residence time in the organ by electrostatic interaction with the cornea, as its distal parts are negatively charged [101].

2′,3′-dideoxyinosine, DDI, was also applied in chitosan-based nanocarriers. It is transported into cells by nucleoside transporters and undergoes phosphorylation to the active triphosphorylated form inhibiting the HIV reverse transcriptase enzyme. DDI, a nucleoside reverse transcriptase inhibitor, has been used against HIV-1 and HIV-2 infections. DDI is transported into cells by nucleoside transporters and then, after entering the cell, it is metabolized by cellular enzymes to its active form, dideoxyadenosine triphosphate (ddATP). The ddATP inhibits the HIV reverse transcriptase enzyme by competing with natural dATP. ddATP inhibits DNA chain elongation by blocking phosphodiester bond formation. DDI is used in clinical practice to treat HIV-1, HIV-2, and other retrovirus infections, including HTLV-1 [1,39,102]. The most toxic effects of DDI are peripheral neuropathies and pancreatitis due to mitochondrial toxicity. The drug is contraindicated in patients with a history of pancreatitis or neuropathy. In addition, cases of peripheral chorioretinal degeneration and optic neuritis have been observed in adults. Concomitant use of other drugs that cause pancreatitis or neuropathy increases the risk and severity of these symptoms. Additionally, clinical cases have shown peripheral chorioretinal degeneration during treatment with DDI in adults [103,104,105].

Chitosan NPs administered intranasally have the ability to penetrate the brain. The idea, with significant findings, is shown in Figure 7. Al-Ghananeem et al. developed an intranasal chitosan-based carrier for DDI, thereby avoiding the inactivation of the drug in the gastrointestinal tract and increasing the therapeutic potential of the drug in brain tissues, which is an essential aspect in the case of therapy against human immunodeficiency virus, due to its occurrence in these tissues [11]. The NPs used in the study revealed a size of 382 nm, loading of 47.3%, and encapsulation efficiency of 94.6%. An in vitro study showed that the NPs were characterized by biphasic release. In the first phase of burst release, 78% of the API was dissolved in the acceptor solution within 10 min. Compared to the intranasal DDI solution, the NPs with DDI were characterized by a greater extent of absorption from the nasal cavity, bioavailability (70.9% and 38.9% for NPs and solution, respectively) and had a higher concentration in brain tissues. In contrast to intravenous administration of DDI, the vehicle had a higher concentration in brain tissues, and the concentration in the blood was much lower in the first thirty minutes and higher after that time. The concentrations of DDI in rat brains at specific time points were 17.16, 22.41, and 42.37 ng/mL for the intravenous solution, intranasal solution, and intranasal NPs, respectively [11].

Narayanan et al. obtained chitosan NPs with TNFA incorporated into an oleogel consisting of ethylcellulose and sesame oil [106]. NPs were prepared by a spray drying technique with the optimized drug-to-polymer ratio and parameters presented in Table 4. The main advantage of this DDS was its prolonged release time—56% of TNFA was released over 16 days. Compared to the NPs alone, and to the oleogel with TNFA, a significant difference in the cumulative API dose released was observed. For NPs containing only TNF, 60% of the dose was transferred to the acceptor medium in less than 4 h, and for the oleogel, after 16 days the smallest percentage of the released dose was 80%. During the tests, the adequate stability of the carrier was confirmed. Ex vivo permeation studies using goat buccal membrane and in vitro cytocompatibility studies on L929 mouse fibroblast lines were also performed. The permeation studies showed that the oleogel with chitosan NPs had the least penetration capacity, which may have been caused by two synergistic barriers: diffusion of TNFA from NPs to the oleogel and diffusion from the oleogel to the membrane. In vitro studies have shown that the use of chitosan NPs in oleogel is safer over free TNFA, as the CC_50′_s were 2.34 mg/mL and 24.98 mg/mL for the free drug and the TNFA NPs loaded in oleogel. Therefore, it may be a promising direction for the development of a pharmaceutical form for TNFA [106].

**Table 4 nanomaterials-13-02647-t004:** Combinations of nanogels with active substances from the purine group.

API	Ø (nm)	ζ (mV)	EE (%)	Summary	Ref.
**6-MP**	187	54.9	n.g.	NPs administered orally lower acute and subacute toxicity than that of plain drug	[92]
	90	26.2	n.g.	formulation development	[107]
	137.9	53.8	29.10	development of NPs administered orally pH-dependent release—95% at pH 4.8 and 75% at pH 7.4, after 24 h slight difference in cytotoxicity between plain drug and the DDS	[93]
**6-TG**	261.63	15.97	44.27	pH-dependent release—57.14% and 19.71% released after 4 h at pH 4.8 and 7.4, respectively blend of 6-TG and curcumin, increased cytotoxicity towards cancer cells (MCF-7, PA-1) curcumin increases the anticancer effect of 6-TG	[23]
**CLA**	636–820	n.g.	62	formulation development of NPs for oral and intravenous administration biphasic release: 38% after 6 h, then an interval of 24–34 h, and further release up to 80 h from the start	[98]
**ACV**	220	4.1	62.5	liver-targeting NPs for intravenous administration lactosaminated NPs—augmentation of drug concentration in hepatic tissue MRT extended from 2.91 to 19.67 h dose recovered from the liver after 8 h increased from 11% to 24% prolonged release in vitro	[6]
	200	36.7	56	ocular delivery of ACV preliminary study—optimization of preparation	[100]
**DDI**	382	n.g.	94.6	intranasal DDI NPs increased API concentrations in brain tissues increase of intranasal bioavailability—38.9% (solution) to 70.9% (NPs) higher concentration of DDI in the brain after intranasal administration compared to intravenous administration	[11]
**TNF**	315	n.g.	49	NPs incorporated into oleogel as long-acting injectable depot system formulation development extended release—56% released in 16 days composition decreased permeability of the drug ex vivo (buccal goat membrane)—may function as a long-acting system decrease in cytotoxicity	[106]
				**To sum up**	
				**Pros**	
				6-MP embedded in chitosan-based NPs lowers acute and subacute toxicity of the drug [92]. 6-TG NPs showed an increased antiproliferative effect when combined with curcumin. Lactosaminated chitosan NPs loaded with ACV increased the drug concentration in hepatic tissues. Intranasal administration of chitosan NPs with DDI influenced a high concentration of the drug in brain [11]. Chitosan-based NPs are promising for DDS for antiproliferative purines, as they are responsive to acidic conditions, with higher drug release, and will be helpful for delivery and release in an acidic tumor environment [23,93].	
				**Cons**	
				Therapeutic efficacy, both in vitro and in vivo, is not known for some prepared DDS [6,92,98,100]. The zeta potential of chitosan is positive, and therefore, it raises safety considerations. Therefore, it might be useful to deepen toxicity evaluations of the DDS, like NP interactions with biological membranes [108].	

### 3.2. Other Nanogels

#### 3.2.1. Gelatin-Based Nanogels

Gelatin is a naturally occurring, biocompatible polymer, basically sourced from animal protein (collagen), that can be obtained by denaturation of the fibrous insoluble collagen from bones, skin, and connective tissue [26,109,110]. A gelatin–water solution reveals its important feature—viscosity. Another critical property of gelatin is its gel strength, known as the Bloom value, which reflects its strength and stiffness. There are two main types of gelatin: type A gelatin with an isoionic point of 6–9, obtained from the acidic hydrolysis of collagen; and type B gelatin (isoionic point of 5) produced from basic hydrolysis. Gelatin of porcine origin is normally referred to as type A gelatin and gelatin of bovine skin origin is referred to as type B gelatin [26].

Scientific studies focused on gelatin in the form of a nanogel as a carrier for active substances from the purine group were carried out in the first decade of the twenty-first century and concerned mainly ACV and DDI. A. Kharia and A. Singhai, in their two works, studied mucoadhesive NPs that contained ACV for oral administration [111,112]. In a research paper from 2014, the authors used a Taguchi standard orthogonal array L_8_ design to describe the influence of process parameters on the physicochemical nature of NPs. They found out that increasing the concentration of the surfactant and cross-linking agent negatively affects mucoadhesion, while a higher concentration of gelatin has a positive impact on it. The dependencies between parameters and factors are shown in Table 5 [111].

In another paper, they described the preparation and evaluation of the nanogel using a central composite design. The most advantageous formulation was characterized by a release of zero-order kinetics and diffusion that deviated from Fick’s laws. The physical parameters of the optimized formulation were as follows: size 217.4 nm, PDI 0.268, EE 70.65%, and drug loading 78.58%. Compared to ACV tablets administered to rats, the vehicle prolonged ACV release, increased bioavailability and MRT, and decreased the maximum concentration, while delaying the time at which this point was observed. To specify, MRT was prolonged from 7.972 h to 12.092 h, and AUC_0–∞_ increased from 3841.13 ng/mL·h to 7527.9 ng/mL·h. In addition, 12 h after administration of the NPs to rats, they were still observed in the gastric mucosa of the animals [112]. Thus, the results prove the possibility of eliminating the pharmacological obstacles of ACV associated with low bioavailability using gelatin nanogels.

Gelatin-based nanogels with DDI were equipped with targeting moieties in order to bind to specific receptors. In the work by Jain et al., gelatin NPs were prepared by a two-step desolvation technique (acetone as desolvating agent and glutaraldehyde as cross-linker), as this technique led to less aggregation of NPs compared to one-step desolvation [113]. Then, the NPs were chemically modified using surface amino acid groups from gelatin and the aldehyde moiety from mannose, thus obtaining the so-called mannosylated NPs. Mannosylation changed the physicochemical parameters of the NPs. The particle size increased from 248 to 325 nm, PDI decreased from 0.439 to 0.324, zeta potential slightly decreased from 10.5 to 6.2 mV, and drug loading reduced from 48.5 to 40.2%. Moreover, this approach allowed for a 2.7-fold increase in DDI uptake by mouse macrophages in relation to non-mannosylated NPs and an 18-fold increase in relation to the uptake of the substance from the drug solution, which was studied ex vivo on a macrophage cell culture (Figure 8). In addition, in vivo studies in mice showed that the NPs quickly reached their maximum concentration at the desired sites of action (macrophage-rich organs such as spleen or lymph nodes), while reducing the concentration of DDI in the systemic circulation [113]. The proposed NPs revealed the potential to eliminate side effects and to increase the therapeutic effect of DDI in the case of HIV infection. A slightly different approach was studied by Kaur et al. [114]. The authors developed gelatin NPs coated with mannan using a double desolvation method followed by coating with polysaccharide. The coating also increased the size (from 120 to 140 nm) and decreased the zeta potential (from 18.6 to 7.2 mV), yet an increase in the entrapment efficiency was observed (from 71.2 to 79.5%). This procedure resulted in a five-fold increase in the uptake of NPs by macrophages and a twelve-fold increase in their accumulation in lymph nodes and the brain compared to the DDI solution, which was supported by ex vivo and in vivo studies on rats. During the microscopic evaluation of macrophages incubated with NPs coated with mannan, a significant change in cell morphology was detected. The macrophages increased in volume, cell nuclei were separated from other cell compartments, and some of the cells underwent lysis. After the addition of a solution of the drug or NPs without mannosylation, the effect on cell morphology was less prominent. In vivo studies on rats revealed that the coated NPs predominantly accumulated in spleen, lymph nodes, and brain, whereas a decrease in the accumulation was noted for kidneys and lungs. Specifically, the accumulation in the brain was 12.4 times higher than for the drug solution and approximately 2 times higher than for non-coated NPs [114]. The nanogel discussed above increased the pharmacological potential of DDI, particularly its permeation into brain tissues, which was beneficial for HIV therapy and had also been studied by other researchers, such as Al-Ghananeem et al. [11].

#### 3.2.2. Poly(ethyleneimine)-Based Nanogels

FLU, a fluorinated analog of the antiviral agent vidarabine, is a purine analog and antimetabolite that inhibits DNA synthesis. FLU is phosphorylated to the nucleoside FLU by deoxycytidine kinase (*dCK*) after entering the cell. Further, in the cell, it is phosphorylated to active FLU triphosphate. The active drug inhibits DNA polymerase, DNA primase, DNA, and RNA ligase, and it is incorporated into DNA and RNA, causing inhibition of RNA processing and mRNA translation, which results in the inhibition of DNA synthesis, and destruction of the cancer cells [1,39]. FLU is indicated for the treatment of adult patients with B-cell chronic lymphocytic leukemia showing a lack of response to therapy or disease progression during treatment with at least one standard regimen containing an alkylating drug [115,116]. The adverse effects of FLU therapy include myelosuppression with lymphopenia, thrombocytopenia, and anemia. In addition, nausea, vomiting, chills, fever, malaise, anorexia, peripheral neuropathy, and weakness have been reported with FLU therapy. CD4+ T-cell depletion during therapy predisposes patients to opportunistic infections. Therefore, pneumonia was observed during therapy with FLU [1,117,118]. Next, high doses of FLU in elderly patients can cause mental status disorders, seizures, optic neuritis, and coma [119].

Vinogradov et al. developed a nanogel based on a copolymer of PEO and PEI for the nucleoside analog FLU 5′-triphosphate [120]. Firstly, PEI-PEO NPs were obtained using a modified emulsification-solvent evaporation method, during which the nucleoside analog was connected to the matrix through amino groups of PEI and the phosphorus group of the nucleoside. During the optimization of the NPs, it was found that the molar ratio of PEO and PEI should be higher than 9, so that the NPs would display a low cytotoxicity and high swelling capacity. In the next step, NPs were conjugated with folic acid in the presence of 1-(3-dimethylaminopropyl)-3-ethylcarbodiimide hydrochloride. Morphologically, the carrier consisted of a condensed polyplex core composed of the nucleoside and PEI, and a PEO envelope with sporadically conjugated folic acid. The most prominent NPs had a PEO/PEI molecular ratio of 12:1 and total conjugation of the folic acid to amino group at the level of 5%. The NPs were characterized by a hydrodynamic diameter of 58 nm and FLU 5′-triphosphate loading of 130 µg/mg of the nanogel. As a result of folic acid modification, the uptake of NPs by MCF-7 cells was increased eight-fold compared to NPs without the modification. Also, transcellular transport (transcytosis) through Caco-2 cell monolayers of the drug was enhanced four times. In addition, the stability of FLU 5′-triphosphate was increased, which retained its active form in 85% after transcytosis, in contrast to the FLU 5′-triphosphate solution, where this substance was mainly located on the outer side of the cell membrane in the dephosphorylated form. In the context of cytotoxicity, it was noted that the manufactured NPs significantly increased cytotoxicity (IC_50_ = 3 μM) compared to unphosphorylated FLU (IC_50_ = 30 μM), which may be a result of the interaction of a positively charged surface of the NPs with cell membranes. A nanogel formulation based on PEI-PEO offers a significant improvement in the delivery of nucleoside analogs [120]. Triphosphates of nucleosides, due to their charge, reveal a negligible ability to penetrate cell membranes and can be transported by endocytosis using these NPs. Therefore, it is possible to administer substances such as FLU in its active form, thus bypassing the mechanisms of resistance of cancer cells associated with the weakening of the metabolic pathways of phosphorylation. In addition to increasing the therapeutic potential, the nanogel offers a reduction in the required therapeutic dose and a reduction in toxicity, which results from the targeted nature of the nanodevice and the release under the influence of low pH in endosomes.

Another article from Vinogradov et al. focused on PEI-based nanogels as carriers for DDI 5′-triphosphate [121]. The authors prepared five nanogels with different modifications of the polymer. In the first two nanogels, disulfide-bridged PEI was conjugated with PEO or PLU (PEO-PEI with spatial distribution, PLU-PEI layered structure), in the third, branched PEI was conjugated with 1,1′-carbonyldiimidazole-activated PEO (bPEI-PEO, layered structure), in the fourth nanogel, the third conjugate was additionally linked with polyamideamine (bPEI-PEO-PAMAM, cationic core–neutral shell structure), and in the fifth, the PEG-PEI nanogel was modified with apolipoprotein E receptor-binding peptide (AP-PEO-PEI), which is a brain-specific peptide vector. Because of PEI’s toxicity, the carriers were equipped with a high content of neutral polymers. The encapsulation of DDI was performed by mixing the nanogels with the drug solution, yielding NPs with a densely loaded core. After drug loading into the nanogels, the sizes of the particles were 1.5 to 2-fold smaller, between 90 and 120 nm. In ex vivo studies on macrophages, the PEO-PEI nanogel loaded with the drug and ATP BODIPY FL displayed a noticeably higher accumulation in the cytoplasm than a free solution. In particular, the NPs were visible in endosomes. Additionally, a flow cytometry study was conducted in order to evaluate the internalization difference between the prepared nanogels. The internalizations from the highest to the lowest were as follows: bPEI-PEO, PEO-PEI, PLU-PEI, and bPEI-PEO-PAMAM. A positive charge and thinner layer of PEO in bPEI-PEO and PEO-PEI were correlated by the authors with higher internalization, as the particle should opsonize more and, thus, be exposed to the immune system. In the next part of the study, the antiviral activity of the NPs against HIV-1 was examined after 2 h pretreatment using monocyte-derived macrophages. The lowest mean EC_90_ values were observed for PLU-PEI (0.6 µg/mL), bPEI-PEO-PAMAM (0.5 µg/mL), and AP-PEO-PEI with an embedded blend of DDI 5′-triphosphate and azidothymidine 5′-triphosphate (1.2 µg/mL), whereas the value for free DDI was 7.0 µg/mL. Preliminary safety studies were also conducted, and the mitochondrial toxicity of the NPs was evaluated by measurement of mtDNA. The third NP type, bPEI-PEO, caused 75% depletion of the nucleic acids, while in the other ones, no depletion was found. Treatment with the peptide-modified carrier containing the blend of DDI and azidothymidine 5′-triphosphates at a dose of 15 µg/mL produced a 60% increase in the mtDNA content, while at a dose of 30 µg/mL, there was a 14% decrease [121].

A summary of the two studies on PEI-based NPs is presented in Table 6. The major advantages of nanogels combined with APIs from the purine group are shown in Figure 9.

### 3.3. Alternative for Nanogels

Alternative systems to nanogels are microgels. These DDSs were loaded with purines such as theophylline and caffeine. The matrix materials used for these microparticles are PVA, methacrylates, latex, and copolymers based on aspartic acid [122,123,124,125]. These systems provide interesting properties such as sustained release [122,125], thermo-responsive properties [123], and high affinity to specific molecules [124]. Therefore, they can be applied as transdermal DDSs [123] and affinity adsorbents [124].

## 4. Dendrimers

Dendrimers are highly branched macromolecules with the potential to be used for biomedical applications. Several dendrimers are toxic owing to their positively charged surfaces. However, the toxicity can be reduced by coating these peripheral cationic groups with carbohydrate residues. PAMAM dendrimers were the first commercialized dendrimers. Their diameters range from 1 nm to 14 nm (or, respectively, from G0 to G10). As the initiator core, either ethylenediamine (EDA), ammonia (NH_3_), or cystamine can be used, providing different numbers of possible branches. The interior generations are made consecutively from N-(2-aminoethyl)acrylamide in a two-step process. The terminal amine groups can be most frequently functionalized with hydroxyl (OH), carboxylic acid (COOH), or conjugation to hydrocarbon chains and PEO. PAMAM dendrimers have been intended for a wide range of biomedical applications thanks to their tunable physicochemical properties. Novel PAMAM dendrimers are becoming more and more popular in the systemic delivery of drugs, genes, and contrast agents. However, their usage is limited due to doubts about the effect of these properties on endocytosis mechanisms and on the cytotoxicity, dependent on dendrimer generation (i.e., size and surface group density), surface chemistry, dosage, and specificity. The ability to control the physicochemical properties of PAMAM dendrimers, such as size and surface functionality, have made them ideal candidates for biomedical purposes [29,126].

One area of concern is dendrimers’ toxicity [29,30,31,32]. Aisina et al. evaluated the coagulofibrinolytic mechanisms of influence of the charge of the PAMAM dendrimers and analyzed changes in the overall plasma hemostatic potential [29]. As these PAMAM dendrimers of low generation were known not to induce aggregation of human platelets in plasma in vitro, the tests were focused on selected hemostatic parameters. It turned out that cationic dendrimers increased prothrombin time, suppressed thrombin generation in plasma, and altered the conformation and coagulability of fibrinogen, whereas anionic dendrimers did not exert such effects. The influence on coagulofibrinolytic mechanisms was enhanced with the rise in generation or if the dendrimer concentration increased [29]. PEI-derived NPs have recently gained much attention and seem promising components for nonviral gene delivery systems; however, they are reported to exhibit high cytotoxicity, probably as a result of inducing mitochondrially mediated apoptosis [31]. Khansarizadeh et al. identified and evaluated PEI–protein interactions within cells [31]. The majority of the identified PEI-interacting proteins, such as shock proteins, glutathione S-transferases, and protein disulfide isomerase, are involved in the cell apoptosis process. The interaction of PEI dendrimers with the identified proteins could provide some explanation of the mechanism of PEI cell toxicity. Moreover, the discovered PEI–protein interactions were associated with various cell processes, which may help design safer polycationic vectors [31]. Oskuee et al., 2018 prepared different complexes of PEI dendrimers with DNA to assess the parameters affecting in vitro cytotoxicity [32]. Strong electrostatic interactions were observed between PEIs and DNA, which resulted in the formation of nano-sized complexes. It turned out that polyplexes in HBG buffer with higher molecular weight and branched forms of PEI had smaller sizes, which can cause higher cell toxicity and DNA damage. Linear 250 kDa PEI was non-toxic, whereas branched PEIs with lower molecular weights revealed toxicity in a concentration-dependent manner. Similarly, the cytotoxic effects of branched PEIs were proportional to the NP/plasmid ratio. The primary mechanism of cell toxicity was apoptosis. The scientists confirmed that cytotoxicity depended on PEI size [32]. Ziemba et al. [30] studied the in vivo toxicity of the three following types of fourth-generation PPI dendrimers: uncoated (PPI-g4) dendrimers and dendrimers in which 25% or 100% of the surface amino groups were coated with maltotriose (PPI-g4-25%m or PPI-g4-100%m). These PPI dendrimers were administered to Wistar rats, whose body weight, food and water consumption, and urine excretion were monitored daily. Unmodified PPI dendrimers brought about changes in rat behavior, a reduction in food and water consumption, and lower body weight gain. In the case of PPI-g4 and PPI-g4-25%m dendrimers, disturbances in the urine and hematological and biochemical profiles were observed, yet the parameters returned to normal during recovery. PPI-g4-100%m had no toxic effect on rats. During prolonged administration of unmodified PPI-g4, several undesirable effects diminished, indicating the appearance of biological counteracting mechanisms. When dendrimer administration was ceased, all disturbances returned to normal levels. The toxicity of PPI dendrimers increased with the dose- and decreased with the sugar-modification degree. The 100% modification of the dendrimer surface made it harmless at every examined dose, indicating that the cationic amino groups are the cause of PPI dendrimer toxicity. However, the same amino groups are often necessary for achieving a therapeutic effect. Therefore, the researchers confirmed the conclusion that attaching sugar residues may be a good solution to retain the activity of dendrimers and reduce their toxicity [30]. In further work, Ziemba et al. [127] focused on the cytotoxicity and genotoxicity of PPI dendrimers. Unmodified or partially modified with maltotriose PPI dendrimers demonstrated a high affinity to DNA due to the cationic groups on their surface. Coating PPI dendrimers with maltotriose appeared to be an efficient method to reduce their genotoxicity and make them better candidates for therapeutic agents or drug carriers. The researchers confirmed the conclusion from their previous experiments that surface modification with sugar residues may be a good solution to retain the activity of dendrimers and eliminate side effects [127].

Dendrimers used as carriers for API from the purine group usually belong to the fourth generation and are made of PPI. So far, dendrimers have been evaluated as carriers for FLU, CLO, 6-MP, and ACV. The most important information about these dendriplexes is presented in Table 7. Gorzkiewicz et al. performed studies on the binding of ATP (model adenosine nucleotide) with PPI dendrimers using isothermal titration calorimetry and zeta potential titration [128]. These studies provided important information in the context of binding adenosine analogs, such as FLU or CLA. It was found that the binding occurs spontaneously between charged parts of molecules (phosphate moieties and amine surface groups of dendrimers), the low enthalpy of the process stabilizes the complex, and the number of bound molecules is positively affected by a lower pH and negatively by modification of the end groups with maltose or maltotriose. Additionally, the ionic strength had no effect on ATP affinity to the dendrimer. The provided information allows the characteristics that the dendrimer should exhibit to be a promising carrier for purine analogs to be determined. The authors suggested that it should be a maltose-modified dendrimer due to its low enthalpy value and the high Gibbs free energy value, as well as the incomplete saturation of the active sites, as saturation causes a decrease in zeta potential and, thus, a decrease in stability. In addition, maltose-modified dendrimers are characterized by lower cytotoxicity and showed higher stability in alkaline environments, so that dendrimer-API complexes will withstand the acidic conditions in endosomes, and drug release will occur in the cytoplasm [128]. These observations were used in subsequent studies by Gorzkiewicz et al. [129,130]. In a paper from 2018, the authors combined the active form of the drug, FLU triphosphate, with previously studied maltose-modified PPI dendrimers [129]. Providing the active substance in this way has the advantage of overcoming the mechanisms of cancer resistance to drugs. One of the forms of resistance is correlated with reduced uptake of substances caused by silencing the expression of membrane receptors responsible for the passage of FLU through the cell membrane and silencing the expression of proteins responsible for its metabolism to triphosphate. Bypassing of this mechanism led to an observed increase in the in vitro cytotoxicity of the vehicle relative to the FLU solution and the lack of a negative effect on the cytotoxicity after adding an inhibitor of the corresponding nucleoside-transporting membrane receptor [129]. In a paper from 2019, Gorzkiewicz et al. [130] enriched the evaluation of FLU triphosphate and additionally combined CLO triphosphate with a previously manufactured dendrimer. The obtained results indicated a decrease in the cytotoxicity of CLO delivered in this way and, thus, a decrease in therapeutic potential, which resulted from the following reasons: the lack of resistance mechanisms that the carrier could overcome, the occurrence of stronger interactions of CLO triphosphate with the dendrimer, and reduced uptake of the drug [130]. It is worth noting that CLO is a second-generation purine nucleoside analog that contains Cl and F atoms in the structure. Inside the cell, CLO is metabolized to the active metabolites 5′-monophosphate and 5′-triphosphate, which inhibits DNA synthesis and repair by competitively inhibiting DNA polymerases. This action reduces the intracellular pool of deoxynucleotide triphosphates and leads to autopotentiation of CLO triphosphate incorporation into DNA, thereby increasing the effectiveness of DNA synthesis inhibition. CLO is approved for the treatment of pediatric patients, from 1 to 21 years old, with relapsed or refractory acute lymphocytic (lymphoblastic) leukemia. The main side effect of CLO is myelosuppression. Other rare side effects accompanying the drug usage include hypotension, tachyphemia, pulmonary edema, organ dysfunction and fever, elevated liver enzymes, and hyperbilirubinemia; nausea, vomiting, diarrhea, hypokalemia, and hypophosphatemia [1,39,131,132,133,134,135,136].

Initial studies on the use of dendrimers as a carrier for 6-MP date back to the beginning of the twenty-first century. They were carried out by Neerman et al. [137,138]. In the first article, the authors focused on the possibility of reducing 6-MPs hepatotoxicity by encapsulation in melamine-based dendrimers. Parenteral administration of the prepared DDS was correlated with a 36% lower alanine aminotransferase value than for the group taking 6-MP alone [137]. In the second article, the possibility of 6-MP encapsulation in PAMAM dendrimers was investigated. The study showed that for one dendrimer molecule of the fourth generation, there were 38 6-MP molecules at a pH of 7 and this decreased slightly with a decrease in pH [138]. Research on the possibility of increasing the therapeutic potential of 6-MP and reducing its hepatotoxicity was also promising for the treatment of autoimmune hepatitis, where due to the impaired functioning of the diseased organ, it is essential to protect it against other potentially damaging factors [139]. The most recent study on the possibility of using dendrimers as carriers for 6-MP was conducted in 2013 by Wang et al. [140]. The fifth-generation PAMAM dendrimers, terminated with hydroxyl or amine groups, released most of the encapsulated 6-MP within 2 h. However, the authors developed and prepared a hybrid system in which a fifth-generation dendrimer terminated with hydroxyl groups was deposited on a gold NP through sulfhydryl groups. This had a decisive influence on the release of the active substance from the vehicle. Release stopped within 4–24 h at 16.6% from the start of the study. However, the addition of a reducing agent, dithiotreitol, in the 24th hour of the test, resulted in the release of 88.5% of the total 6-MP from the carrier within the next 5 h [140]. Research on the prolongation of 6-MP release is important for the treatment of autoimmune diseases. Replacing the nitroimidazole derivative of 6-MP (azathioprine) by a nanocarrier with 6-MP is potentially beneficial because it can provide the desired effect, prolonged residence time in the body, with a simultaneous lack of side effects, thus bringing the formulation closer to a drug with ideal characteristics.

Yandrapu et al. researched the effect of combining ACV with thiolated PAMAM dendrimers [141]. The plain dendrimers were conjugated with cysteamine and then loaded with ACV. After the modification and loading, the size of the dendrimers increased from 5.6 nm to 173 nm, and the zeta potential increased slightly from −23.1 mV to −19.9 mV. Moreover, the thiolated derivatives showed improved mucoadhesion (rat small intestine), slowed down release, and reduced the ACV content in the carrier. The mucoadhesion was stronger in a formulation where a higher amount of thiolating agent was used. It was concluded that a decrease in ACV concentration has no significant effect due to the possibility of compensation by the administration of more dendrimers [141]. The study is part of the trend in formulation development in which pharmacokinetic parameters are improved through mucoadhesion in the gastrointestinal tract.

The advantages of combining purines with dendrimers are shown in Figure 10.

**Table 7 nanomaterials-13-02647-t007:** Combinations of dendrimers with active substances from the purine group.

API	Dendrimer Material	Generation	Summary	Ref.
**FLU**	poly(propyleneimine)	fourth	dendrimers modified with maltose improved pharmacokinetics bypassing FLU resistance mechanisms increasing the therapeutic potential	[129,130]
**CLO**	poly(propyleneimine)	fourth	dendrimers modified with maltose reduction of therapeutic potential	[130]
**6-MP**	melanin	fourth	reduced hepatotoxicity of the drug loaded in dendrimers and delivered via intraperitoneal injection	[137]
	hydroxyl terminated poly(amidoamine)	fourth	encapsulation efficiency research per 1 molecule of dendrimer there were 38 molecules of 6-MP encapsulation decreases with a decrease in pH	[138]
	hydroxyl terminated poly(amidoamine) with gold NPs ⌀ 3 nm	fifth	release dependent on the concentration of the reducing agent prolongation of release	[140]
**ACV**	thiolated poly(amidoamine)	modified third	NPs for oral delivery increased retention in the digestive system—mucoadhesion through thiolation of the dendrimer	[141]
			**To sum up**	
			**Pros**	
			Maltose-derived dendrimers are less toxic than non-modified ones. Maltosylated PPI dendrimers showed increased antiproliferative potential in vitro when combined with FLU, as it bypassed one of the resistance mechanisms [129,130]. 6-MP in dendrimers revealed reduced hepatotoxicity and prolonged drug release [137,138,140] which might be helpful in treatment of autoimmune diseases and act as a substitution for azathioprine. Thiolation of PAMAM dendrimers loaded with ACV enhanced mucoadhesion in GI tract [141]. Modification of dendrimers with saccharides reduced toxicity in vitro and in vivo [30,127].	
			**Cons**	
			Maltosylated PPI dendrimers combined with CLO did not increase the therapeutic potential of the drug, as there were no resistance mechanisms to bypass [130]. The more PEI dendrimers are branched the higher the observed toxicity [32]. Dendrimers have multistep synthesis procedure, which makes them expensive for large scale production [31]. Some dendrimers reveal toxicity [29,30,31,32], thus additional studies need to be considered in order to increase their applicability in pharmaceutical technology.	

## 5. Polymeric Micelles with Purines

Polymer micelles were studied in connection with 6-MP, 6-TG, and ACV. It is worth noting that controlled-release nanocarriers were the most often prepared for 6-MP. Literature reports on the formulation of purine analogs with polymeric micelles are combined in Table 8. Zhang et al. chemically bound 6-MP through a disulfide bond to a branched, biodegradable polymer molecule consisting of a hydrophobic main chain (pyrrolidone-2,5-dione mers) and hydrophilic branching of PEO (Figure 11(1)) [142]. The polymeric chains self-assembled in solution with 6-MP directed to the center of the globular micelle and PEO directed outside, yielding NPs 160 nm in size and with a zeta potential of −12.4 mV. The disulfide bridge connection resulted in a release dependent on the presence of reducing agents such as dithiothreitol; 35.2% and 71.1% of the dose was released from the nanocarrier within 85 h in a solution without reducing agent and from 10 mM dithiothreitol, respectively. In addition, cytotoxicity studies on leukemic cells (HL-60) showed that the vehicle is characterized by lower cytotoxicity in the concentration range of 0.5–500 µg/mL compared to the free drug at the same concentration range [142]. Thus, the “nanodevice” was characterized by adequate biocompatibility. The reduction in the cytotoxicity of the NPs compared to the free drug may be due to prolonged release, which translates into the incomplete release of 6-MP during a 48 h cytotoxicity test. Shi et al. also composed NPs that released 6-MP in the presence of a reducing agent [143]. The sulfhydryl group of the active substance was modified using prop-2-yonic acid and then combined via a carboxyl group methylated at one of the ends of the PEO (mPEG-1000 or mPEG-5000) chain (Figure 11(2)). The synthesized materials, PM-PEG-1000 and PM-PEG-5000, were able to self-assemble into micelles with CMCs of 0.0045 and 0.0272 mg/mL, respectively, while the diameters, PDIs, and zeta potentials were 55.8 nm, 0.102, and −3.0 mV for PM-PEG-1000 and 72.6 nm, 0.116, and −1.4 mV for PM-PEG-5000. PM-PEG-1000 presented better uptake by the monocyte-macrophage cell line J774 compared to the second formulation. The unsaturated bond in the linker was susceptible to nucleophilic attack and able to undergo a Michael’s addition–elimination reaction, therefore, it was used for 6-MP release. This feature was observed during in vitro release studies, where 12.1% of the dose was released in a 100 μM glutathione solution and 53.7% and 77.8% in 2 mM and 10 mM solutions of PM-PEG-1000, respectively. Moreover, when the glutathione concentration was increased, an increase in the diameter of the micelles was also observed, which is a result of a decrease in the hydrophobic nature of the polymer chains caused by the release of 6-MP and substitution for glutathione in the chain. In cytotoxicity studies on HL-60 cells, a lower toxic potential of PM-PEG-1000 was noted compared to that of the free drug, which may result from reduced NP uptake by HL-60 cells. Studies carried out in this field showed that 6-MP from NPs was captured to a lesser extent by HL-60 cells—17.5 μg compared to 19.6 μg for the control group (drug solution). Moreover, the NPs were characterized by adequate stability under storage conditions, and PEO contributed to an appropriate character, reducing opsonization and uptake by macrophages [143]. Liao et al. also developed a polymeric carrier for 6-MP that released API as a result of a change in pH and a reducing agent [144]. 6-MP was modified using N-(prop-2-yn-1-yl)prop-2-ynamide and connected to a suitable moiety in the polymeric matrix, azide. In addition, tertiary amine and doxorubicin were linked to the matrix (Figure 11(3)). The material was able to self-assemble into micelles of 116 nm in size. By introducing the amino moiety, the negative surface charge (−7.29 mV) was changed to positive (9.31 mV) at pH 6.5. The micelles revealed adequate stability in fluids that mimicked the physiological and intercellular environment—at pH 7.4 and at micromolar glutathione concentrations, 17% of doxorubicin and 20% of 6-MP were released after 24 h. In contrast, in a fluid mimicking intracellular conditions, the micelles increased in diameter and released 37% and 75% of doxorubicin and 6-MP, respectively. In vitro toxicity studies showed lower cytotoxicity of the NPs compared to free solutions of the active substances, which was due to a slower release from the micelles and NP penetration into the cell. The IC_50_ for free 6-MP was calculated to be 2.17 and 3.21 μM against HeLa and HL-60 cells, respectively, while in NPs, the values were 3.97 and 4.13 μM, respectively. Moreover, the cytotoxic effect was pH-dependent: the IC_50_ values (micelles with doxorubicin and 6-MP) were 0.87 and 0.73 μM at pH 7.4 and 6.5, respectively, against HeLa cells [144].

Kaur et al. prepared nanocarriers based on vinylbenzylthymine, its N-methylated form, and vinylbenzyltriethylammonium chloride, in which 6-MP and 6-TG were enclosed (Figure 11(4)) [145]. The synthesis of the material was performed by a TEMPO-mediated living radical polymerization system in H_2_O/ethane-1,2-diol. The highest encapsulation values were observed for polymers in which thymine was not methylated on a nitrogen atom. In addition, the amount of active substance in the NPs were influenced by the degree of cross-linking—the higher it was, the smaller the encapsulation. Also, the size of the NPs was the smallest in the formulation with the core-cross-linked polymer: 69.9 and 70.5 nm for NPs with 6-MP and 6-TG, respectively. Moreover, the authors conducted experiments on the release of drugs from carriers using a pressure-driven membrane process. For this purpose, an ultrafiltration membrane was assembled. This choice resulted from the best reconstruction of the actual kinetics of release of the active substance from the carrier. Unmethylated NPs slowed the release of both 6-TG and 6-MP, which can be explained by the occurrence of additional interactions—hydrogen bonds. In addition, it was concluded from the experiments that the higher the degree of cross-linking of the material and the larger the diameter of the micelles, the less API passes to the acceptor solution from the nanocarrier [145]. Jeanbart et al. developed micelles based on PEO and poly(propylene sulfide), to which 6-TG was linked through a disulfide bridge (Figure 11(5)) [146]. The NPs were studied for their ability to kill myeloid suppressor cells (MDSCs), responsible for the deterioration of the immune system response to tumors in the body. Compared to the 6-TG solution, micelles significantly reduced the concentration of MDSC in blood of mice and increased the efficacy of adoptively transferred T cells. Moreover, after therapy, there was no recurrence of the disease, while in the group that did not receive 6-TG, recurrence appeared at a level of 25%; thus, this approach increased the potential for cancer immunotherapy [146]. Zheng et al. developed carboxymethyl chitosan NPs with 6-MP conjugated to the polymer matrix through a disulfide linker [147]. The conjugation of hydrophobic 6-MP allowed for self-assembly of the material to NPs, as the 6-MP part formed a hydrophobic core and the polysaccharide part formed a hydrophilic shell. The obtained NPs were monodispersed and spherical, with diameters of 155.8 nm (DLS) and 100 nm (TEM). The NPs were resistant to the release of 6-MP for up to 3 days in solutions without glutathione or with 2 μM of the compound. Moreover, in a solution containing 100 μM of glutathione, less than 10% of the 6-MP dose was released from the NPs for up to 3 days. Conversely, for concentrations of 2 mM and 10 mM glutathione in the acceptor solution, 65.1% of the total 6-MP amount was released from the NPs within 3.5 h and 74.4% in 3 h, respectively. These results indicate that in the intercellular environment containing micromolar glutathione concentration, excessive drug release will not occur, although it may appear in the cytosol, in which the glutathione concentration is within the range of 2–10 mM. In addition, the release was influenced by pH, lower values reduced the release rate, which was due to tighter packing of the NPs and a lower content of the anionic form of glutathione [147]. The developed DDS may be useful in anticancer therapies due to the increased glutathione level in their cells [148].

The advantages of micelle-based delivery of 6-MP and 6-TG are shown in Figure 12.

Sawdon and Peng focused on the development of micelles for purines with antiviral activity—ACV and GCV [149,150]. The materials prepared in these studies are shown in Figure 13. In an article from 2014, ACV was linked to poly(caprolactone) through an ester bond, and next the hydrophilic part of PEO or chitosan was connected in order to yield an amphiphilic copolymer. The obtained micelles had a different physicochemical character. Micelles with a PEO chain had a lower hydrodynamic diameter (141.8 compared to 172.7 nm), lower zeta potential (1.4 compared to 32.3 mV), and faster release. However, the release mode was similar—biphasic with a stage of initial burst release lasting 2 h and releasing approximately 50% of the active substance. After 48 h, the total release was 96% for micelles with the PEO part and 82% for micelles with the chitosan part. In contrast, during cytotoxicity studies on HT29 cells, no significant differences in toxicity between NPs were detected and the results indicated good biocompatibility [149]. In an article from 2015, Sawdon and Peng connected GCV to chitosan and poly(caprolactone)-based micelles [150]. The size of the hydrophobic core and the zeta potential were 117 nm and 24.2 mV, respectively. The obtained vehicle was characterized by prolonged release, dependent on hydrolysis and diffusion, with an initial rapid release phase. This phase was most likely caused by a weakening of the ester bond between GCV and the NPs when chitosan was connecting to polycaprolactone. Studies conducted on the HT29 cell line showed adequate biocompatibility, a small beneficial change in cell survival, but no changes in growth morphology were observed [150].

**Table 8 nanomaterials-13-02647-t008:** Polymeric micelles in combination with active substances from the purine group.

**API**	**Ø** **(nm)**	**Summary**	**Ref.**
**6-MP**	160	6-MP conjugated to pyrrolidone-2,5-dione mers through disulfide bond release of the drug depends on the presence of a reducing agent lower toxicity to leukemia cells HL-60	[142]
	55.8	drug conjugated to PEO through prop-2-yonic acid release of the drug depends on the presence of a reducing agent micelles with shorter PEO chains were more intensively uptaken by phagocytes	[143]
	116	nanocarrier with 6-MP and DOX addition of tertiary amine changed zeta potential of NP to positive release depends on the presence of a reducing agent and pH lower cytotoxicity than that of free drug	[144]
	115.8	6-MP conjugated to the matrix through a pH-sensitive linker glutathione-dependent release; negligible up to 100 uM, 65.1%, and 74.7% after 3.5 h in the GSH concentrations of 2 mM and 10 mM, respectively	[147]
	163.3	micelles based on vinylbenzyl derivatives carriers with methylated thymine were characterized by lower EE% and faster release cross-linking of the polymer caused decrease in diameter of NPs	[145]
**6-TG**	162.4		
	30	micelles based on PEO and poly(propylene sulfide) for injection 6-TG conjugate to the matrix through a disulfide bond reduces the concentration of MDSC in mice blood increases the efficacy of adoptively transferred T cells smaller recurrence rate	[146]
**ACV**	141.8	carrier-based on poly(caprolactone) and PEO ACV conjugated through an ester bond slower release than NPs based on poly(caprolactone) and chitosan 96% of drug released in 48 h	[149]
	172.7	NPs based on poly(caprolactone) and chitosan ACV conjugated through an ester bond 82% of drug released after 48 h	
**GCV**	117	NPs based on poly(caprolactone) and chitosan drug release depends on hydrolysis and diffusion from the carrier biphasic release—during conjugation of PEO, some of the bonds between drug and polymer weakened	[150]
		**To sum up**	
		**Pros**	
		6-MP and 6-TG were chemically bound to polymeric matrix through pH- or glutathione-sensitive linkers, which gave them the potential to release at the tumor site [142,143,144,146,147]. Micelles based on PEO and poly(propylene sulfide) reduced concentration of MDSC in circulatory system, reduced recurrence rate of malignancy, and increased efficacy of adoptively transferred T cells [146]. Chemical conjugation of ACV and GCV to polymeric matrix, like poly(caprolactone), have the potential to slow drug release [149,150].	
		**Cons**	
		There is a need for stronger evidence of better antiproliferative effect of micelles with 6-MP or 6-TG conjugated through pH- or glutathione-responsive bonds. The usefulness of ACV and GCV chemically bound to micelles in antiviral therapies is still to be explored.	

## 6. Conclusions and Perspectives

Many purine derivatives are APIs in drug products of significant importance in the therapy of autoimmune diseases, cancers, and viral infections. However, in many cases, their medical use is limited due to certain unfavorable physicochemical and pharmacokinetic properties. Common issues that negatively affect purine effectiveness in therapy are a short half-life, low bioavailability, short interval of drug dosing regimen, low concentration, and short residence time at the site of action as well as toxicity. These problems were partially overcome by the preparation of new purine analogs in the form of prodrugs. Modern drug delivery systems and the use of NPs, including polymer NPs, create prospects for solving the API limitations.

Polymeric NPs have a generally positive effect on the therapeutic potential of purine APIs. The use of polymeric NP-purine systems results in improved pharmacokinetic properties (such as AUC, t_1/2_, MRT, clearance) as compared to the APIs used alone or in commercially available formulations. The described DDSs offer protection for the embedded APIs in many ways, including shielding them from the unfavorable environment of the gastrointestinal tract, thus significantly improving their bioavailability when oral dosing is concerned, as well as protecting them from metabolism and factors in the body (i.e., enzymes) that would trigger their premature activation outside of the target site, degradation, or elimination. Also, when an appropriate DDS is used, modified release is often achieved in an intranasal formulation, enabling the API to cross the blood–brain barrier.

Regarding the biological activity of the delivered purines, purine APIs were often found to express lower IC_50_ values when combined with polymeric NPs, which was translated in in vivo animal models to prolonged survival of animals. This was due to a higher concentration of the APIs in the desired target location in the body, tissue, or cell, owing to the polymeric carrier, which additionally led to less profound side effects. All these findings are encouraging in terms of their potential introduction into clinical practice. However, some knowledge gaps must be filled to avoid certain dangers.

Many purine analogs have not been tested at all, or their evaluation has been limited to a certain DDS. Some of the studies do not concern the biological evaluation of the formulations. This fact only shows that the general reported idea is feasible but not whether a DDS could be used for a broader spectrum of APIs. Some of the tested delivery systems exert a cytotoxic effect by themselves, which raises concerns about the safety of their use or the potential development of drug resistance toward the carrier itself. Appropriate types of DDS improved therapeutic outcomes; however, in some cases, the effect was worse than the use of the drug alone. This indicates that each DDS-API formulation should be considered individually and significant emphasis should be put on the safety of the formulation, including carriers that exhibit a positive charge in particular. Nevertheless, as the polymeric NPs were found capable of improving the biological properties of purine APIs, the topic should be further pursued. In this review, concerns were raised regarding blanks and dangers associated with the therapeutic systems that should be considered when studying new DDSs. Based on the current state of knowledge, polymeric NPs should be further studied in the direction of targeted delivery, both in terms of specific organ targeting (e.g., liver or brain tissue) and cellular localization (pH-responsive DDS).

To sum up, there is still a need to improve the therapeutic efficacy of the APIs from the purine group, and in this regard, the utilization of drug delivery systems in the form of NPs seems to be especially promising. Recent progress in polymer-based NPs with purine-active pharmaceutical ingredients has provided an opportunity to eliminate their disadvantages through modifications of unfavorable physicochemical and toxicological properties, pharmacokinetics, and bioavailability. Therefore, in vitro and in vivo biological studies of these systems are of immense value for the determination of their efficacy, which can increase therapeutic applications.

## Figures and Tables

**Figure 1 nanomaterials-13-02647-f001:**
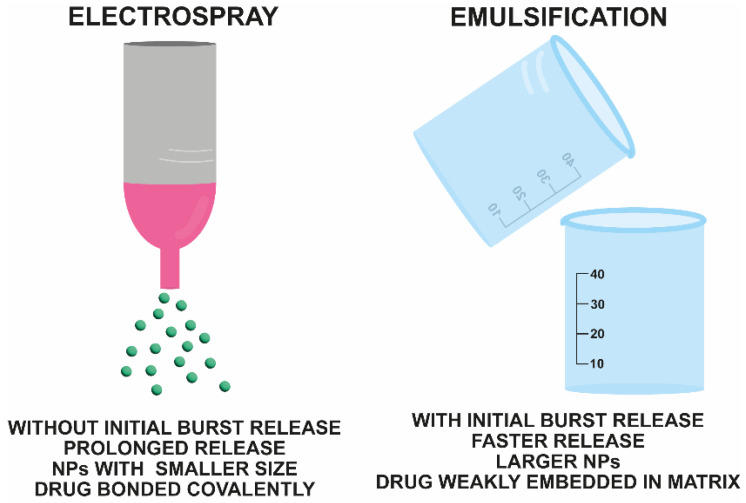
Comparison of electrospray and emulsification methods; based on Chatterjee et al. [47].

**Figure 2 nanomaterials-13-02647-f002:**
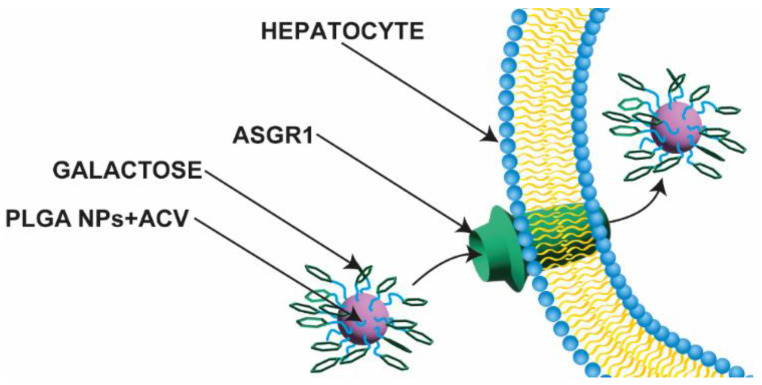
Receptor-mediated uptake of NPs, prepared by Gupta et al. [9].

**Figure 3 nanomaterials-13-02647-f003:**
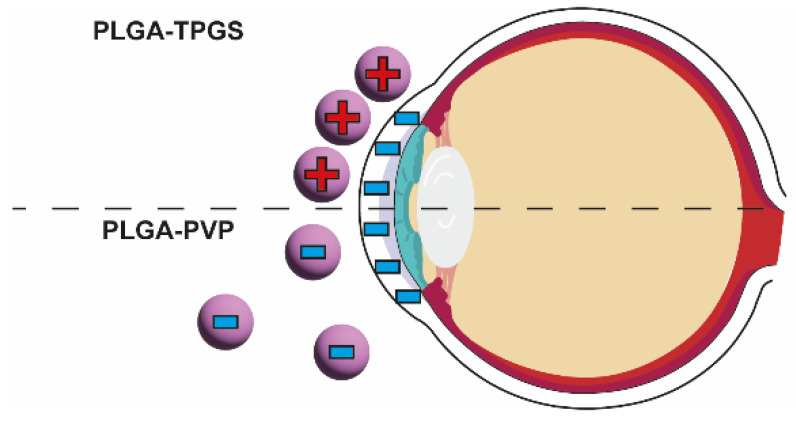
Comparison of adherence to eyeball of PLGA-TPGS and PLGA-PVP NPs loaded with ACV; based on data presented by Alkholief et al. [7]. TPGS—tocopheryl polyethylene glycol succinate, PVP—poly(vinylpyrrolidone).

**Figure 4 nanomaterials-13-02647-f004:**
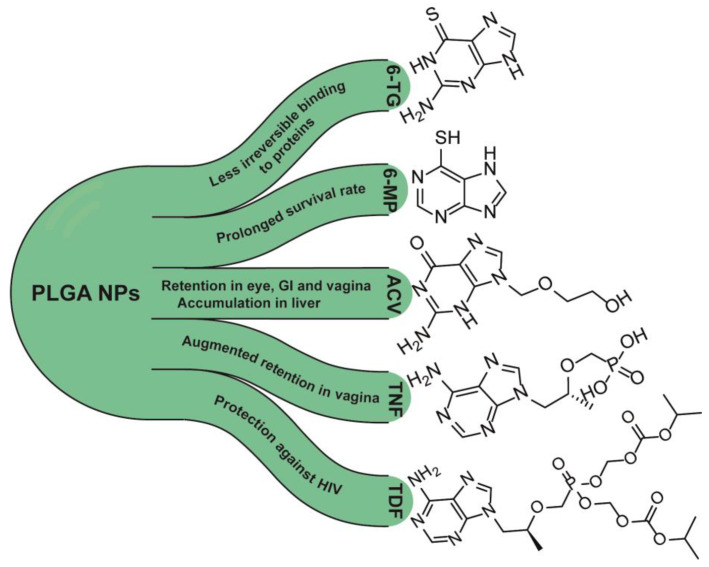
Major milestones achieved in PLGA-NPs-purines combination.

**Figure 5 nanomaterials-13-02647-f005:**
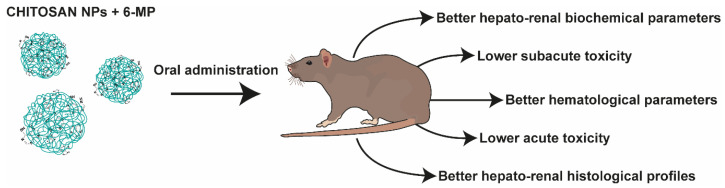
Chitosan nanoparticles as carriers for purines.

**Figure 6 nanomaterials-13-02647-f006:**
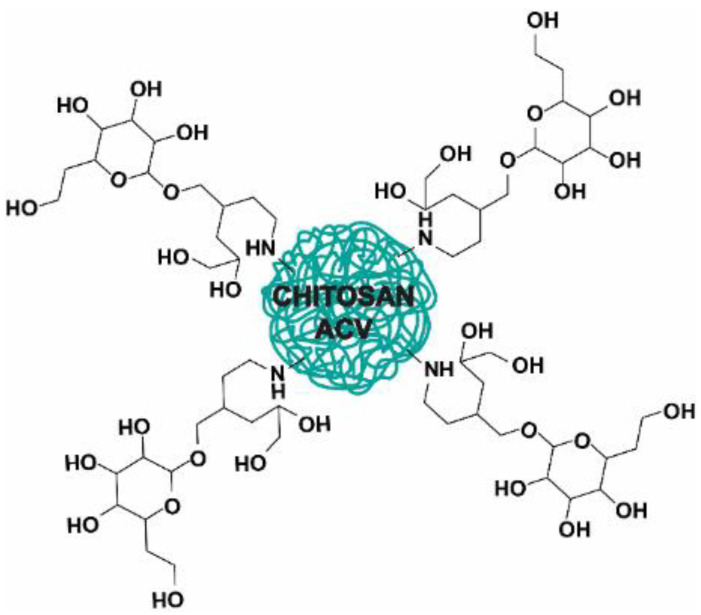
Lactosylated chitosan NPs with ACV, prepared by Jain et al. [6].

**Figure 7 nanomaterials-13-02647-f007:**
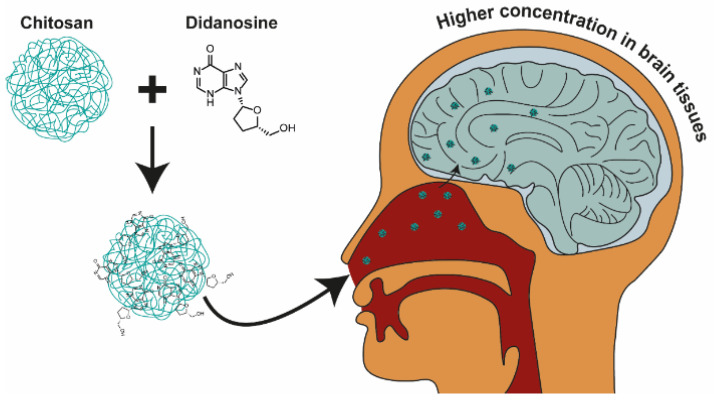
A possible formulation and route of administration for DDI as an anti-HIV agent [11].

**Figure 8 nanomaterials-13-02647-f008:**
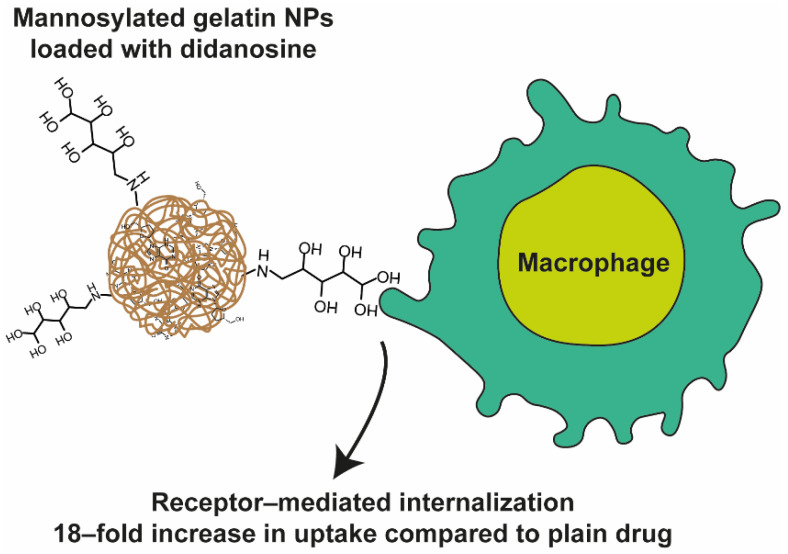
Mannosylated gelatin NPs prepared by Jain et al. [113].

**Figure 9 nanomaterials-13-02647-f009:**
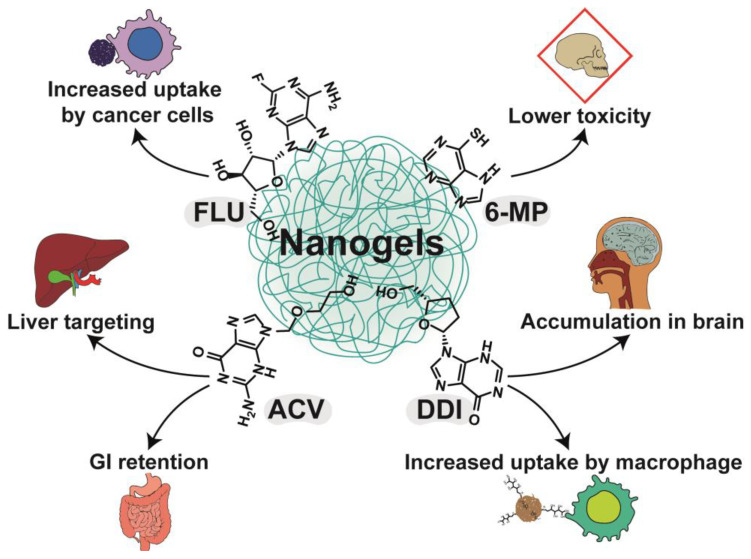
Major advantages of nanogels combined with APIs from the purine group.

**Figure 10 nanomaterials-13-02647-f010:**
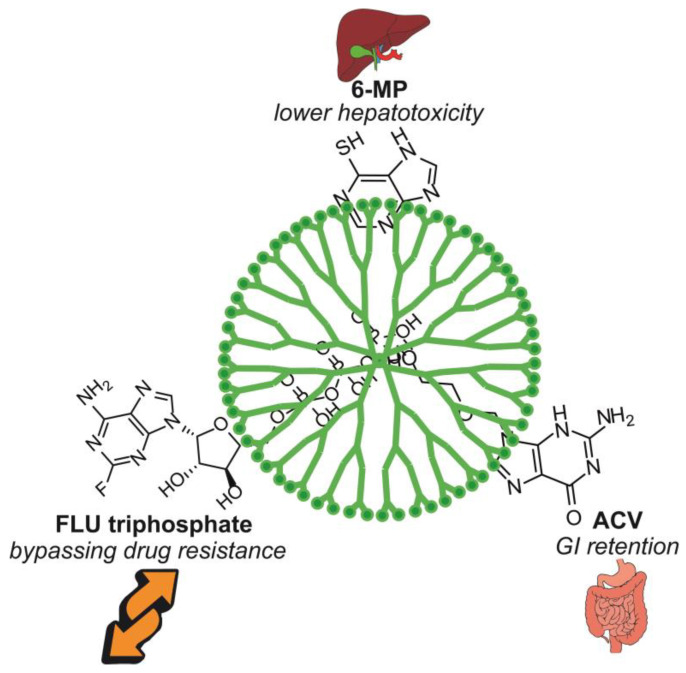
Advantages of purines combined with dendrimers.

**Figure 11 nanomaterials-13-02647-f011:**
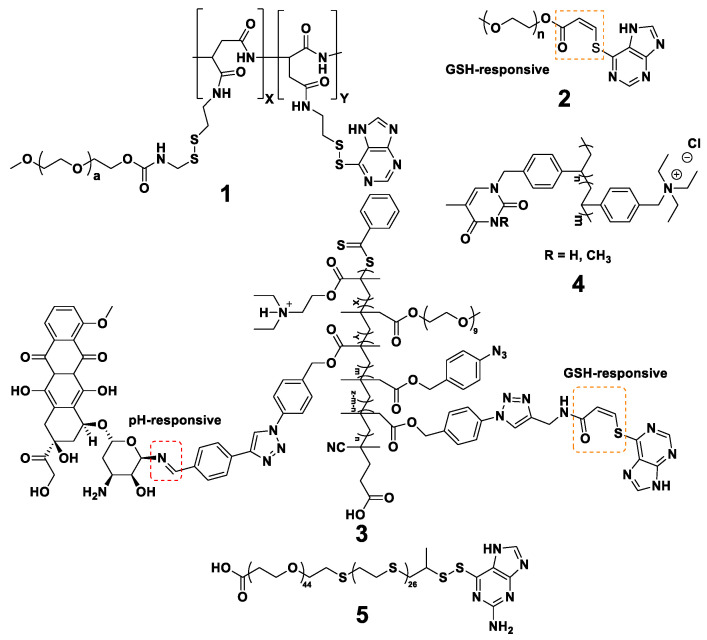
Different polymers used for the preparation of micelles: polymeric prodrug of 6-MP (**1**) prepared by Zhang et al. [142]; polymeric prodrug of 6-MP (**2**) synthesized by Shi et al. [143]; polymeric prodrug of doxorubicin and 6-MP (**3**) synthesized by Liao et al. [144]; polymer matrix synthesized for micelles with embedded 6-MP and 6-TG (**4**) by Kaur et al. [145]; polymeric prodrug of 6-TG (**5**) synthesized by Jeanbart et al. [146].

**Figure 12 nanomaterials-13-02647-f012:**
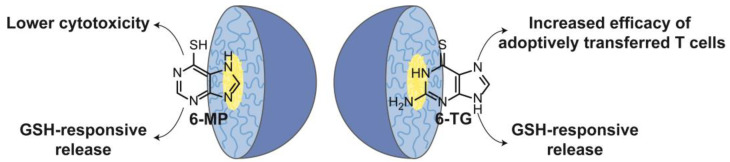
Comparison of milestones for 6-MP and 6-TG combined with polymeric micelles.

**Figure 13 nanomaterials-13-02647-f013:**
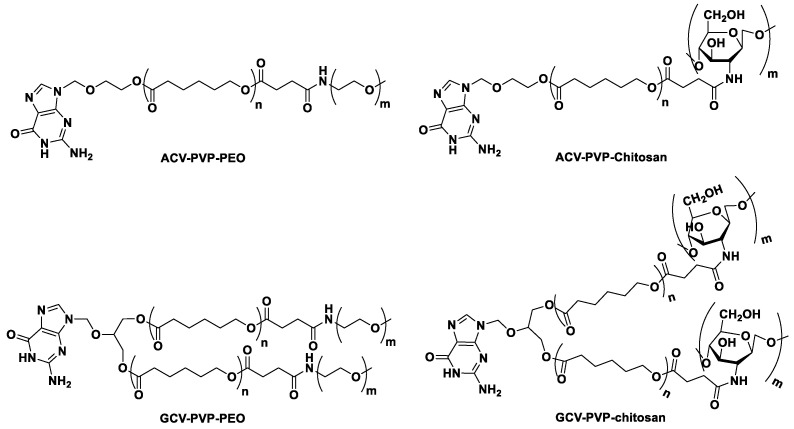
Different polymeric prodrugs of ACV and ACV studied by Sawdon and Peng [149,150].

**Table 1 nanomaterials-13-02647-t001:** Comparison of the more promising formulation and commercially available drug with ACV studied by Kamel et al. [58]; AUC_0–∞_—area under the curve, MRT—mean residence time.

Formulation	C_max_ (µg/mL)	AUC_0–∞_ (μg/mL·h)	t_1/2_ (h)	Clearance (L/h)	MRT (h)
Commercial	26.37	38.52	1.34	0.5284	2.11
PLGA-ACV	23.15	803.34	45.10	0.0252	63.22

**Table 2 nanomaterials-13-02647-t002:** Benefits of using purines in PLGA nanoparticles.

API	Ø (nm)	ζ (mV)	EE (%)	Summary	Ref.
**6-MP**	138.01	−1.0	80.71	oral PLGA NPs core–shell structure drug release faster in a more alkaline environment bimodal release—initial burst release followed by a sustained release lower inhibitory impact on Jurkat T cells of NPs than that of plain drug, yet increased rate of apoptosis of Jurkat T cells when treated with NPs more favorable pharmacokinetics of NPs prolonged median survival time of in vivo ALL model mice—from 23.5 to 51 days	[46]
**6-TG**	149.10	−36.6	97.22	6-TG conjugated chemically to PLGA supported by an electric field simultaneous conjugation and preparation of NPs elimination of burst release extended prolonged release of NPs prepared by electrospraying compared to emulsification—60–65% in 30 days compared to approx. 92% in 14 days weaker irreversible binding of the drug to serum proteins higher IC_50_ value of the 6-TG-NPs than that of the free drug	[47]
**ACV**	285	−10.1	14.02	formulation development of parenteral NPs bimodal release—burst followed by prolonged NPs with surfactants (Tween 80, PVA, PLU F-127) and ACV had less negative zeta potential additional coating by PLU resulted in a decrease in negative zeta potential, an increase in size, and faster drug release the best formulation was prepared from lactic acid:glycolic acid 75:25 and a 1% solution of PLU F-127 as surfactant, and without coating NPs had more favorable pharmacokinetics in vivo reduced risk of thrombophlebitis	[58]
	198.1	−8.5	61.10	PLGA conjugated with galactose for intravenous administration galactosylation increased NP size and reduced cumulative release both NPs, galactosylated and not, showed biphasic release of the drug increased liver accumulation of the drug reduction in hemolysis improvement in pharmacokinetics	[9]
	262	15.14	74.12	ocular (topical) delivery system synthetic vitamin E was more beneficial over polyvinylpyrrolidone in terms of adherence to eyeball parameters of the DDS were comparable to eye fluids adequate safety in vivo on rabbits increased drug retention at the administration site—24 h post-administration, the concentration was in the therapeutic range	[7]
	164.9	n.g.	59.7	ocular (topical) delivery system thermosensitive gel elimination of rapid release when NPs were embedded in the gel use of dipeptide derivatives to target a specific receptor to transport the drug the best ligand for the receptor had two L-valines in the structure and had faster bioreversion to ACV	[8]
	740	n.g.	54.4	mucoadhesive NPs with polycarbophil for oral delivery increased retention in GI tract increased bioavailability	[60]
	400	28.73	80.16	buccal films with beneficial pharmacokinetics	[63]
	161	−6.7	n.g.	vaginal NPs penetrating mucus at administration site increased retention in vagina HSV protection	[10]
**TNF**	275	−29.5	n.g.	for pre-exposure therapy vaginal administration of NPs embedded in film leakage of the drug from the formulation was low	[73]
	127	48.4	n.g.	vaginal administration for pre-exposure therapy increased bioadhesiveness to pig vaginal mucosa increase in TNF concentration in NPs—the problem of low loading capacity was solved	[74]
	300–700	>0	n.g.	NPs for intravenous administration PLGA-chitosan NPs did not modify API release	[76]
**TDF**	336.8	−2.38	21.9	vaginal administration NPs with the addition of methacrylate copolymers for pre-exposure therapy pH-dependent release—semen responsive	[72]
	148.6	−26.7	24	NPs for vaginal delivery NPs protected against HIV-1 infection	[75]
	218	−4.8	52.9	NPs for oral administration protection of the prodrug against degradation and improvement of pharmacokinetics	[77]
				**To sum up**	
				**Pros**	
				PLGA NPs showed good biocompatibility/lowered toxicity of treatment in many routes of administration—ocular [7], parenteral [9,58], oral [46], and vaginal [72]. Administration of 6-MP embedded in PLGA NPs revealed a prolonged survival rate of ALL model mice [46]. Burst release of 6-TG from PLGA NPs may be prevented by electrospraying technique, which is beneficial for formulations with prolonged release [47]. PLGA NPs reveal higher solubility and release drugs faster in a more alkaline environment, which is beneficial for the prevention of viral diseases and treatment of leukemia [46,72]. For PLGA-ACV NPs, enhanced retention in the eye [7] and vagina [10] was noted, and augmented concentration in the liver was measured when NPs were galactosylated [9]. Combination of PLGA with polycarbophil increases retention in GI and translates to higher bioavailability [60]. PLGA NPs with TDF, TNF, and ACV are promising formulations for vaginal pre-exposure prophylaxis as they enhance penetration through vaginal mucosa, retention, and safety [10,72,73,74,75]. Opsonization and uptake by the immune system can be used to deliver anti-HIV agents to macrophages [76].	
				**Cons**	
				Vaginal leakage in some cases is still relatively high and further development is needed [73,75]. The positive zeta potential of some NPs may raise safety concerns. Little is known about the formulation stability of PLGA NPs combined with purine derivatives, especially those with zeta potential above −30 mV. TNF is problematic to load into PLGA NPs, and additional excipients should be added [74].	

**Table 3 nanomaterials-13-02647-t003:** Combination of polymethacrylate in combination with active substances from the purine group.

API	Ø (nm)	ζ (mV)	EE (%)	Summary	Ref.
**ACV**	236	n.g.	79.34	formulation development over 75% of the dose released in 24 h	[20]
	99	26.1	n.g.	gel with NPs for the prevention of sexually transmitted HSV administered intravaginally improved pharmacokinetics and retention in vagina	[84]

**Table 5 nanomaterials-13-02647-t005:** Effect of different factors on parameters of gelatin nanoparticles from the work of A. Kharia and A. Singhai [111]. Symbols: ↑—increase, ↓—decrease, ↑*—increase up to gelatin pI, *—the highest near ACV pI, n.g.—not given, PDI—polydispersity index.

	Gelatin ↑	Stabilizer ↑	Cross-Linking Agent ↑	Acetone ↑	pH ↑	Stirring Speed ↑	Stirring Time ↑
**Size**	n.g.	↓	↓	n.g.	↑	↓	n.g.
**PDI**	↓	n.g.	↑	n.g.	↑*	↓	n.g.
**Loading efficiency**	n.g.	↑	n.g.	↑	*	↓	↑
**Drug release**	↓	↓	n.g.	n.g.	n.g.	n.g.	n.g.
**Mucoadhesion**	↑	↓	↓	n.g	n.g	n.g	n.g

**Table 6 nanomaterials-13-02647-t006:** Gelatin and poly(ethyleneimine)-based nanogels in combinations with active substances from the purine group.

API	Material	Ø (nm)	ζ (mV)	EE (%)	Summary	Ref.
**ACV**	Gelatin	165–1610	n.g.	39–80	formulation development of orally delivered NPs optimization of NPs by Taguchi standard orthogonal array L_8_ design gelatin increased mucoadhesion, while PLU and glutaraldehyde decreased it	[111]
	Gelatin	274.4	n.g.	70.65	formulation development of orally delivered NPs optimization of NPs using a central composite design mucoadhesive NPs improved pharmacokinetic parameters in rats retention in the gastric tract was observed	[112]
**DDI**	Gelatin	325	6.2	n.a.	mannosylated NPs for intravenous administration increased uptake by macrophages mannosylation decreased positive zeta potential accumulation in macrophage-rich organs—spleen, lymph nodes	[113]
	Gelatin	140	7.2	79.5	mannan-coated NPs for subcutaneous injection increased uptake by macrophages increased permeation through the blood–brain barrier accumulation in macrophage-rich organs decreased concentration of drug in the lungs and kidneys	[114]
	PEI	90–120	n.g.	n.g.	five nanogels with different compositions PEO-PEI with spatial distribution PLU-PEI layered structure star bPEI-PEO layered structure, most toxic towards mitochondria bPEI-PEO-PAMAM, cationic core–neutral shell, best antiviral activity AP-PEO-PEI—with brain-specific peptide vector addition of drugs caused decrease in particle size	[121]
**FLU**	PEI	58	n.g.	130 µg/mg	PEI conjugated with PEO and folic acid delivery of the active form of the drug, 5′-triphosphate addition of drugs caused a decrease in particle size—condensed polyplex core and PEO envelope PEI to PEO at ratio above 9 is less toxic uptake was increased eight-fold in MCF-7 compared to NPs without folic acid	[120]
					**To sum up**	
					**Pros**	
					Gelatin-based NPs have gastroretentive properties which may be utilized for oral administration of ACV in order to improve pharmacokinetics [111,112]. Coating of gelatin NPs with mannan or mannose increased drug uptake and accumulation in macrophage-rich organs. Therefore, they have the potential as DDS for anti-HIV agents like DDI [113,114]. PEI is an interesting material from the chemical point of view. The macromolecules can be linear, branched, or dendrimeric. Modifications of the material have an impact on NP arrangement, activity, and toxicity. bPEI-PEO-PAMAM loaded with DDI is preferential for further evaluation as it has the most antiviral activity among tested NPs by Vinogradov et al. [121]. The design of experiments is a useful tool in NP optimization as it gives correlation between factors and parameters useful for formulation development. The antiviral potential of NPs loaded with DDI and decorated with mannan or mannose is still unknown and future studies are worth consideration.	
					**Cons**	
					PEI-based NPs showed considerable toxicity. Especially bPEI-PEO layered structure and NPs with PEO:PEI ratio below 9 [120,121].	

## Data Availability

Not applicable.

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
