# Peer review of "Polymer-Based Nanoparticles as Drug Delivery Systems for Purines of Established Importance in Medicine†"

_nanomaterials, 2023, doi:10.3390/nano13192647_

Round 1

Reviewer 1 Report

Overall the authors have done a careful work and the presentation of results is clear. There are however quite a few points that require the author's attention before the paper can be accepted for publication.

1.     I would suggest the author to make the review more compact. The author should explain previously reported literature in a compact way by mentioning the main key problem and the key findings, not all the details.

2.     Cite a few references in support of  lines 38-42.

3. The conclusion should be a little bit short and future perspective should be more specific  and elaborate in terms of different problems with the current reported system.

Author Response

Dear Reviewer, thank you for your insightful comments. My revision was added as an attachment. Best regards Tomasz Goslinski

Reviewer 2 Report

The manuscript "Polymer-based nanoparticles as drug delivery systems for purines of established importance in medicine" presents a review about polymeric drug delivery systems for purines as drugs. There are several polymeric carriers described (PLGA, Chitosan, Gelatine, PAMAM/PPI/PEI and others). The different particles are compared on the basis of a literature search. Several tables summarize the features (in numbers: size, zeta potential, encapsulation efficiency and properties). The similar particles are summarized with their pros and cons below the main tables. Finally, all particles are summarized in the conclusion and perspectives are given. There are some minor points that I would like to raise before the article can be published. The article is very well written and summarizes all features quite well. So it clearly deserves publication after minor changes.

(*) There are recent developments of microgels. They sometimes also have stimuli responsiveness. I am not sure about purins if they can be encapsulated. Otherwise it would be worthwhile to mention these particles as an option as well.

(*) The preferred and the possible administrations do not always become very clear. I fear not all cited references give an idea about that - but I haven't checked. However, I would very much like a quick summary about the administration possibilities in each case of polymeric particles.

Author Response

(The authors gave the same response as above.)
